# DIN-SQL: Decomposed In-Context Learning of Text-to-SQL with Self-Correction

**Mohammadreza Pourreza**
Department of Computer Science
University of Alberta
Edmonton, CA
`pourreza@ualberta.ca`

**Davood Rafiei**
Department of Computer Science
University of Alberta
Edmonton, CA
`drafiei@ualberta.ca`

## Abstract

There is currently a significant gap between the performance of fine-tuned models and prompting approaches using Large Language Models (LLMs) on the challenging task of text-to-SQL, as evaluated on datasets such as Spider. To improve the performance of LLMs in the reasoning process, we study how decomposing the task into smaller sub-tasks can be effective. In particular, we show that breaking down the generation problem into sub-problems and feeding the solutions of those sub-problems into LLMs can be an effective approach for significantly improving their performance. Our experiments with three LLMs show that this approach consistently improves their simple few-shot performance by roughly 10%, pushing the accuracy of LLMs towards SOTA or surpassing it. On the holdout test set of Spider, the SOTA, in terms of execution accuracy, was 79.9 and the new SOTA at the time of this writing using our approach is 85.3. Our approach with in-context learning beats many heavily fine-tuned models by at least 5%. Additionally, when evaluated on the BIRD benchmark, our approach achieved an execution accuracy of 55.9%, setting a new SOTA on its holdout test set.

## 1 Introduction

Natural language interfaces to databases aim at making it easier for end users to access data in a relational database. For example, given the utterance "find employees who make more than their managers" and the schema of tables *employees* and *manages*, one may want to generate a query in SQL that retrieves those employees from a database. Over the past two decades, research in this field has progressed through several phases, with early systems being domain-specific, supporting controlled natural language [Popescu et al., 2003, 2004, Li et al., 2007, Li and Jagadish, 2014] or relying on rule-based approaches [Stratica et al., 2005] while more recent systems offering greater domain-independence using supervised models trained on diverse domains and datasets [Zhong et al., 2017, Yu et al., 2018] and more recently deep neural models trained on large text and code repositories [Dong and Lapata, 2016, Devlin et al., 2018].

The latest development in this progression is the use of Large Language Models (LLMs) under zero-shot and few-shot prompting [Rajkumar et al., 2022, Liu et al., 2023a]. It has been shown that LLMs provide strong baselines using only a few demonstrations and no fine-tuning [Chen et al., 2021, Brown et al., 2020, Liu et al., 2023b]. However, these models fall behind on commonly used benchmarks (e.g., Spider) compared to well-designed and fine-tuned models. Table 1 shows the performance of two latest LLMs, CodeX and GPT-4, on the development set of the Spider dataset. Despite a strong performance, LLMs fall behind, compared to existing methods [Scholak et al., 2021, Li et al., 2023a], especially on medium and complex queries. The question investigated in this paper

37th Conference on Neural Information Processing Systems (NeurIPS 2023).

is where these LLMs fail and if some of the problems that they are facing can be mitigated to push the performance to reach or surpass fine-tuned SOTA models.

Prompting has several advantages over traditional approaches using pretraining or fine-tuning. The main benefit is that LLMs can perform prediction tasks without requiring large task-specific training data. Training models from scratch or fine-tuning them is a resource-intensive process, often requiring a large number of training samples and machine resources, which may not be available. Additionally, few-shot prompting has been shown to outperform previous state-of-the-art methods on several benchmark datasets and can achieve high accuracy even with limited training examples [Brown et al., 2020, Wei et al., 2022b].

It has been recently shown that the performance of LLMs can be improved on more complex tasks (e.g., math word problems, compositional navigation steps) using approaches such as chain-of-thought [Wei et al., 2022b], least-to-most [Zhou et al., 2022], and decomposed [Khot et al., 2022] prompting techniques where a task is broken down into multiple steps and the intermediate results are used to generate a final answer. Unlike algebraic expressions, which consist of clear steps or operations, breaking a complex SQL query can be a more daunting task because of the declarative structure of the language and the complex relationships between query clauses.

In this paper, we propose a novel method based on few-shot prompting that decomposes the task of natural language text to SQL (referred to as text-to-SQL) into multiple sub-tasks. Previous works on text-to-SQL prompting using LLMs are only evaluated in a zero-shot setting [Rajkumar et al., 2022, Liu et al., 2023a]. However, zero-shot prompting only provides a lower bound on the potential power of LLMs for most tasks [Zhang et al., 2022, Kojima et al., 2022, Wei et al., 2022b, 2021, Brown et al., 2020]. We show that our proposed method outperforms the few-shot prompting method by a large margin. We also

| Fine-tuning approaches | |
|---|---|
| **Method** | **Execution accuracy** |
| RED-SQL 3B + NatSQL [Li et al., 2023a] | 84.5 |
| T5-3B + PICARD [Scholak et al., 2021] | 79.3 |

| Inference-only approaches | |
|---|---|
| **Method** | **Execution accuracy** |
| Zero-shot GPT-4 (Ours) | 64.9 |
| Few-shot GPT-4 (Ours) | 67.4 |
| Zero-shot CodeX [Rajkumar et al., 2022] | 55.1 |
| Few-shot CodeX (Ours) | 61.5 |

Table 1: Zero-shot and few-shot prompting compared to fine-tuned approaches on the dev set of Spider

compare our method with previous approaches on two cross-domain challenging benchmarks, Spider and BIRD. For Spider dataset, we use the two official evaluation metrics of *execution accuracy* and *exact set match accuracy* [Zhong et al., 2020]. We utilize two variants of the CodeX family, namely Davinci and Cushman [Chen et al., 2021], and the GPT-4 model for prompting. On the holdout test set of Spider, our method achieves an execution accuracy of 85.3% and 78.2% respectively using GPT-4 and CodeX Davinci models and an exact set match accuracy of 60% and 57% respectively using the same models. The large gap between the exact match and execution accuracies is due to the few-shot in-context nature of our method. Pretrained and fine-tuned approaches are more likely to generate SQL queries with a higher exact set match accuracy simply because these models have seen many examples during training that follow the composition style of the queries in the test set (queries in both sets are often written by the same people). Before our work, the SOTA on the test set had an execution accuracy of 79.9% [Li et al., 2023a] and an exact set match accuracy of 74% [Li et al., 2023b], and our method sets a new ground in terms of the execution accuracy. On the BIRD benchmark, our approach achieves a new SOTA result, attaining an execution accuracy of 55.9% on the holdout test set and 50.72% on the development set when employing GPT-4. Moreover, using the *valid efficiency score* introduced in this benchmark, our approach outperformed a GPT-4 baseline, demonstrating a 9% improvement on the development set. This highlights the effectiveness of our method.

Our contributions can be summarized as follows: (1) improving the performance of LLM-based text-to-SQL models through task decomposition, (2) introducing adaptive prompting strategies tailored to task complexity, (3) addressing schema linking challenges in the context of prompting, and (4) using LLMs for self correction. To replicate the reported results, visit our GitHub repository [1] for access to the prompts, results, and the code.

---

[1] https://github.com/MohammadrezaPourreza/Few-shot-NL2SQL-with-prompting

## 2   Related Work

Sequence-to-sequence models [Sutskever et al., 2014] have shown great potential in code generation tasks including text-to-SQL. The key idea is to jointly encode a given natural language question and the database schema and leverage a decoder to predict the target SQL.

On the encoder side, learning a representation for the question and the database schema is carried out using bidirectional LSTM in IRNet [Graves and Graves, 2012], convolutional neural networks in RYANSQL [Choi et al., 2021], pretrained language models such as BERT in SQLova [Hwang et al., 2019] and graph neural networks in RATSQL [Wang et al., 2019], SADGA [Cai et al., 2021], and LGESQL [Cao et al., 2021]. Gan et al. [2021] propose an intermediate representation to bridge the gap between the natural language question and SQL statements. There has been also work on tabular language models that encode both tables and text such as TaBERT [Yin et al., 2020], TaPas [Herzig et al., 2020], and Grappa [Yu et al., 2020].

The methods on the decoder side can be categorized into sketch-based slot-filling and generation-based methods [Qin et al., 2022]. Sketch-based methods break the problem into several slot prediction sub-problems and aggregate the predictions for the slots of the SQL query to be generated [Hwang et al., 2019, Xu et al., 2017, Hui et al., 2021]. A drawback of these methods is that they cannot generalize to queries that do not follow the predefined templates. The generation-based methods [Guo et al., 2019, Wang et al., 2019, Cao et al., 2021, Huang et al., 2021] decode the SQL query as an abstract syntax tree.

In contrast to pretrained and fine-tuned models, Rajkumar et al. [2022] and Liu et al. [2023a] conduct an evaluation of the zero-shot prompting capability of LLMs on text-to-SQL using different prompts on the Spider dataset. Prompting techniques have been also used for tasks such as table understanding, table reasoning, and table-to-text generation [Guo et al., 2023, Chen, 2022], and some remarkable results have been reported using LLMs with just a small number of examples given in the prompt.

## 3   Few-shot Error Analysis

To better understand where LLMs fail under a few-shot setting, we randomly sampled 500 queries from different databases in the training set of the Spider dataset, excluding all databases used in our prompts. We searched for the queries that produced results different than those of gold queries, hence failing the execution accuracy. We manually examined these failures and classified them into six categories as shown in Figure 1 and discussed next.

**Schema linking** This category contained the largest number of failed queries and included instances where the model failed to identify column names, table names, or entities mentioned in questions. In some cases, the query required an aggregation function, but a matching column name was chosen instead. For instance, the database schema for question "What are the average and maximum capacities for all stadiums?" included a column named "average", which was selected by the model instead of taking the average of the capacity column.

**JOIN** This was the second largest category and included queries that needed a JOIN but the model was unable to identify all the tables required or the correct foreign keys to join the tables.

**GROUP BY** This category included cases where the SQL statement required a GROUP BY clause, but the model either did not recognize the need for grouping or wrong columns were used for grouping the results.

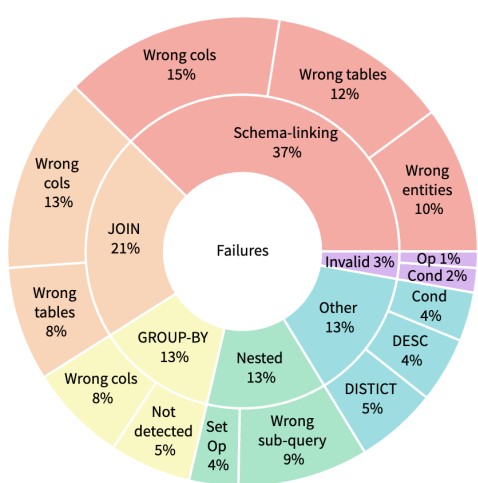

Figure 1: Statistics of simple few-shot failures using CodeX Davinci (Op refers to operators, Cond refers to conditions, and cols refers to columns)

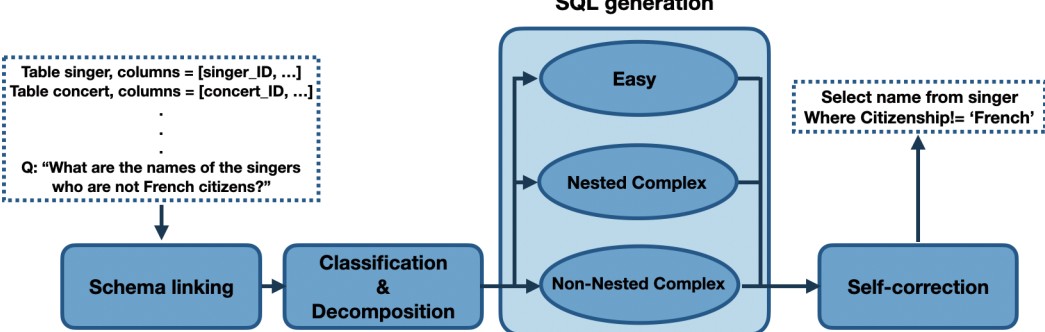

Figure 2: An overview of the proposed methodology including all four modules

**Queries with nesting and set operations**
For this category, the gold query used nesting or set operations but the model did not recognize the nested structure or was unable to detect the correct nesting or set operation.

**Invalid SQL** A small set of the generated SQL statements had syntax errors and could not be executed.

**Miscellaneous** This category included cases that did not fit under any of the previously mentioned categories. Examples included SQL queries that contained extra predicates, missed a predicate, or had missing or redundant DISTINCT or DESC keywords. This category also included cases where the WHERE clause was missing or the query had redundant aggregation functions.

## 4  Methodology

Despite improvements over zero-shot, few-shot models struggle on more complex queries including those where schema linking is less trivial and the queries that use multiple joins or have a nested structure, as discussed in Section 3.

Our approach to address these challenges is to break down the problem into smaller sub-problems, solve each sub-problem, and use those solutions to construct a solution for the original problem. Similar approaches (e.g., chain-of-thought prompting [Wei et al., 2022b] and least-to-most prompting [Zhou et al., 2022]) have been taken to improve the performance of LLMs on tasks that can be broken down into multiple steps such as math word problems and compositional generalization [Cobbe et al., 2021, Lake and Baroni, 2018]. Unlike these domains where the tasks have a procedural structure with one step directly feeding into the next step, SQL queries in most parts are declarative and the possible steps and their boundaries are less clear. However, the thought process for writing SQL queries may be broken down to (1) detecting database tables and columns that are relevant to the query, (2) identifying the general query structure for more complex queries (e.g., group by, nesting, multiple joins, set operations, etc.), (3) formulating any procedural sub-components if they can be identified, and (4) writing the final query based on the solutions of the sub-problems.

Based on this thought process, our proposed method for decomposing a text-to-SQL task consists of four modules (as depicted in Figure 2): (1) schema linking, (2) query classification and decomposition, (3) SQL generation, and (4) self-correction, which are explained in detail in the following sub-sections. While these modules may be implemented using techniques from the literature, we implement them all using prompting techniques to show that LLMs are capable of solving them all if the problems are simply broken down to the right level of granularity. The few-shot examples used in the prompts are obtained from the training set of the respective benchmarks.

### 4.1  Schema Linking Module

Schema linking is responsible for identifying references to database schema and condition values in natural language queries. It is shown to help with the generalizability across domains and the synthesis of complex queries [Lei et al., 2020], making it a critical preliminary step in almost all

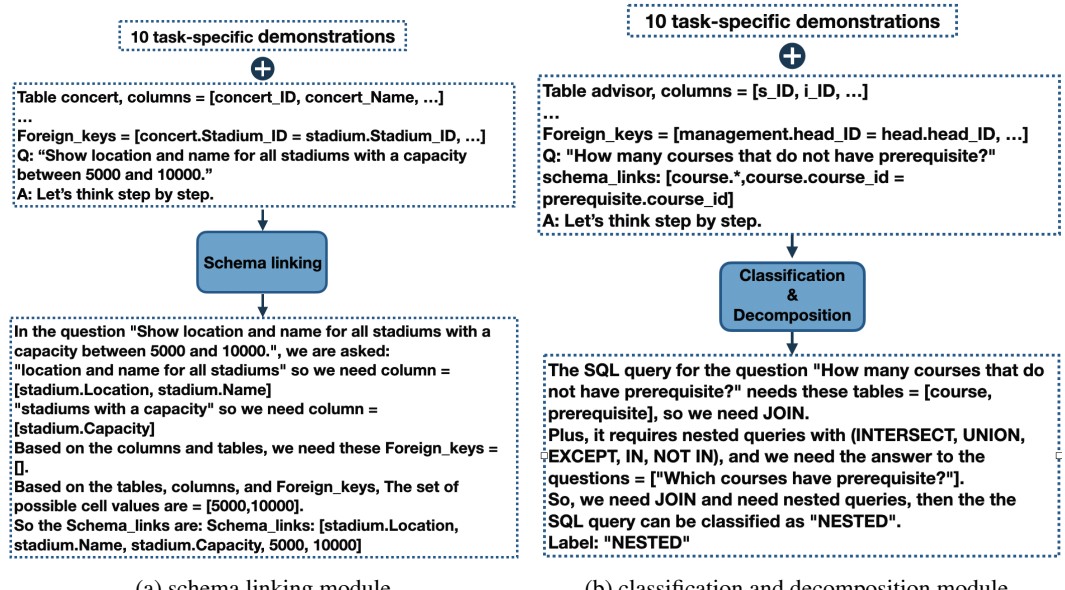

(a) schema linking module        (b) classification and decomposition module

Figure 3: Examples showing the input and output of schema linking (left) and classification and decomposition (right)

existing text-to-SQL methods [Cao et al., 2021, Wang et al., 2019, Guo et al., 2019, Xuan et al., 2021]. This was also a single category with the largest number of failures made by the LLM in our case (Figure 2).

We designed a prompt-based module for schema linking. The prompt includes ten randomly selected samples from the training set of the Spider dataset. Following the chain-of-thought template [Wei et al., 2022b], the prompt begins with "Let's think step by step," as suggested by Kojima et al. [2022]. For each mention of a column name in the question, the corresponding columns and their tables are selected from the given database schema. Possible entities and cell values are also extracted from the question. Figure 3a illustrates an example and the full prompt can be found in Appendix A.3.

## 4.2 Classification & Decomposition Module

For each join, there is some chance that a correct table or join condition is not detected. As the number of joins in a query increases, the chance that at least one join fails to generate correctly increases. One way to alleviate the problem is introduce a module that detects the tables to be joined. Also some queries have procedural components such as uncorrelated sub-queries, which may be generated independently and be merged with the main query.

To address these issues, we introduce a query classification and decomposition module. The module classifies each query into one of the three classes: easy, non-nested complex and nested complex. The easy class includes single-table queries that can be answered without join or nesting. The non-nested class includes queries that require join but no sub-queries, and the queries in the nested class can contain joins, sub-queries and set operations. The class labels are important for our query generation module, which uses different prompts for each query class. In addition to class labels, query classification and decomposition also detects the set of tables to be joined for both non-nested and nested queries as well as any sub-queries that may be detected for nested queries. Figure 3b shows an example input given to the model and the output that the model generates.

## 4.3 SQL Generation Module

As the queries become more complex, additional intermediate steps must be incorporated to bridge the gap between the natural language question and the SQL statement. This gap, known as the *mismatch problem* in the literature [Guo et al., 2019], poses a significant challenge to SQL generation, which stems from the fact that SQL is primarily designed for querying relational databases and not

representing the meaning in natural language [Kate, 2008]. While more complex queries can benefit from listing the intermediate steps in a chain-of-thought style prompting, such listings can degrade the performance for simpler tasks [Wei et al., 2022b]. On the same basis, our query generation comprises of three modules, each geared toward different classes.

For questions in our *easy class*, a simple few-shot prompting with no intermediate steps is adequate. The demonstration for an example $E_j$ of this class follows the format <$Q_j, S_j, A_j$>, where $Q_j$ and $A_j$ give the query text in English and SQL respectively and $S_j$ indicates the schema links.

Our *non-nested complex* class includes queries that require join. Our error analysis (§ 3) revealed that finding the right columns and foreign keys to join two tables can be challenging for LLMs under simple few-shot prompting, especially when the query requires joining multiple tables. To address this issue, we resort to an intermediate representation to bridge the gap between queries and SQL statements. Various intermediate representations have been introduced in the literature. In particular, SemQL [Guo et al., 2019] removes operators JOIN ON, FROM, and GROUP BY, which have no clear counterparts in natural language queries, and merges the HAVING and WHERE clauses. NatSQL [Gan et al., 2021] builds upon SemQL and removes the set operators. Expressions in natural language queries may not clearly map to a unique SQL clause or they may map to multiple clauses, so removing operators makes the transition from natural language to SQL easier. As our intermediate representation, we use NatSQL, which is shown to have a state-of-the-art performance when combined with other models [Li et al., 2023a]. The demonstration for an example $E_j$ of the non-nested complex class follows the format <$Q_j, S_j, I_j, A_j$>, where $S_j$ and $I_j$ respectively denote the schema links and the intermediate representation for the jth example.

Lastly, the nested complex class is the most sophisticated type and requires several intermediate steps before generating the final answer. This class can contain queries that not only require sub-queries using nesting and set operations such as EXCEPT, UNION, and INTERSECT but also multiple table joins, same as the previous class. To break down the problem further into multiple steps, our prompt for this class is designed in a way that the LLM should first solve the sub-queries, generated from the previous module, and then use them to generate the final answer. The prompt for this class follows the format <$Q_j, S_j$ , <$Q_{j_1}, A_{j_1}, ..., Q_{j_k}, A_{j_k}$> , $I_j, A_j$>, where $k$ denotes the number of sub-questions, and $Q_{j_i}$ and $A_{j_i}$ respectively denote the $i$-th sub-question and the $i$-th sub-query. As before, $Q_j$ and $A_j$ denote the query in English and SQL respectively, $S_j$ gives the schema links and $I_j$ is a NatSQL intermediate representation.

Full prompts for all three query classes are provided in Appendix A.4, and all examples for the three classes are obtained from the exact same training set database chosen for the classification prompt.

### 4.4 Self-correction Module

The generated SQL queries can sometimes have missing or redundant keywords such as DESC, DISTINCT and aggregation functions. Our experience with multiple LLMs indicates that these issues are less common in larger LLMs (e.g., queries generated by GPT-4 have less bugs than those from CodeX) but are still present. To address this, we propose a self-correction module where the model is instructed to correct those minor mistakes. This is achieved in a zero-shot setting, where only the buggy code is provided to the model and it is asked to fix the bugs. We propose two different prompts for the self-correction module: *generic* and *gentle*. With a generic prompt, we request the model to identify and correct the errors in the "BUGGY SQL". The gentle prompt, on the other hand, does not assume the SQL query is buggy, and instead asks the model to check for any potential issues and provides some hints on the clauses to be checked. Our evaluation indicates that a generic prompt can yield a better result with the CodeX model, while a gentle prompt is more effective for the GPT-4 model. Unless explicitly stated otherwise, the default self-correction prompt in DIN-SQL is set to gentle for GPT-4 and generic for CodeX. Examples of both generic and gentle self-correction prompts can be found in Appendix A.6.

## 5 Experiments

### 5.1 Models

We evaluated the proposed method using two variants of the CodeX family (Davinci and Cushman variants) and the GPT-4 model. These are the largest open-access LLMs at the time of writing this

paper. Smaller models are less applicable since prompting is believed to be an emergent ability of the LLMs with the number of parameters in the scale of billions [Wei et al., 2022a].

## 5.2 Hyperparameter

All models were accessed via the OpenAI API. Greedy decoding was used to generate the output by setting the temperature at zero. The max tokens was set to 350 for the self-correction module and 600 for all other modules. The stopping token sequence was set to "#;\n \n" for the self-correction module and "Q:" for all other modules.

## 5.3 Dataset

Our evaluation was conducted on two cross-domain challenging datasets, Spider and BIRD. Spider consists of 10,181 questions and 5,693 unique complex SQL queries across 200 databases, covering 138 domains, each containing multiple tables. The standard protocol for this dataset divides it into 8,659 training examples across 146 databases, 1,034 development examples across 20 databases, and a holdout of 2,147 test examples across 34 databases. The databases used in each of these sets are non-overlapping. SQL queries are categorized into four difficulty levels, based on the number of SQL keywords used, the presence of nested subqueries, and the usage of column selections and aggregations. BIRD comprises an extensive dataset with 12,751 unique question-SQL pairs, encompassing 95 large databases totaling 33.4 GB in size. It spans a wide array of more than 37 professional domains, including blockchain, hockey, healthcare, and education. BIRD also introduces external knowledge as an additional resource to assist models in generating accurate SQL queries. Specifically four sources of external knowledge were introduced: numeric reasoning knowledge, domain knowledge, synonym knowledge, and value illustration. Notably, the SQL queries in the BIRD dataset tend to be more intricate than those in the Spider dataset. Language models without access to database content often encounter challenges with schema linking. Therefore, our prompts for the BIRD dataset include sample rows from each table to aid the model in schema linking. Furthermore, we have concatenated the provided external knowledge for each question as a hint, placed immediately after each question. However, due to constraints such as limited context window size, the presence of external knowledge, and the inclusion of sample rows, we have had to reduce the number of demonstrations within the prompts for the BIRD dataset.

| Model | EX | EM |
|---|---|---|
| DIN-SQL + GPT-4 (Ours) | **85.3** | 60 |
| RESDSQL-3B + NatSQL (DB content used) [Li et al., 2023a] | 79.9 | 72 |
| DIN-SQL + CodeX davinci (Ours) | 78.2 | 57 |
| Graphix-3B+PICARD (DB content used) [Li et al., 2023b] | 77.6 | **74** |
| SHiP+PICARD (DB content used) [Zhao et al., 2022] | 76.6 | 73.1 |
| N-best Rerankers + PICARD (DB content used) [Zeng et al., 2022] | 75.9 | 72.2 |
| RASAT+PICARD (DB content used) [Qi et al., 2022] | 75.5 | 70.9 |
| T5-3B+PICARD (DB content used) [Scholak et al., 2021] | 75.1 | 71.9 |
| RATSQL+GAP+NatSQL (DB content used) [Gan et al., 2021] | 73.3 | 68.7 |
| RYANSQL v2 + BERT [Choi et al., 2021] | - | 60.6 |
| SmBoP + BART [Rubin and Berant, 2020] | - | 60.5 |

Table 2: Execution accuracy (EX) and exact set match accuracy (EM) on the holdout test set of Spider

## 5.4 Metrics

The performance of our models are evaluated using the official metrics of each dataset: exact-set-match accuracy (EM) and execution accuracy (EX) for Spider and valid efficiency score (VES) and execution accuracy (EX) for BIRD.

The exact-set-match accuracy (EM) treats each clause as a set and compares the prediction for each clause to its corresponding clause in the reference query. A predicted SQL query is considered correct only if all of its components match the ground truth. This metric does not take values into account. The execution accuracy (EX) compares the execution output of the predicted SQL query with that of the ground truth SQL query on some database instances. Execution accuracy provides a more precise estimate of the model's performance since there may be multiple valid SQL queries

for a given question, and exact set match accuracy only evaluates the predicted SQL against one of them. The Valid Efficiency Score (VES) is a metric designed to measure the efficiency of running the generated SQL queries. This metric is meaningful if the generated queries are correct, meaning their result matches that of the reference query. Therefore, the VES metric takes into account both the accuracy of the generated queries and their efficiency in terms of the execution time.

## 5.5 Results

### 5.5.1 Test set results

As shown in Table 2 for the holdout test set of Spider, our method achieves the highest execution accuracy using GPT-4 and the third-highest execution accuracy using CodeX Davinci among all officially published results at the time of this writing. This is achieved without even utilizing the database content. In terms of exact set match accuracy, our method achieves comparable results to previous works that do not utilize database content. As demonstrated in Table 3a, in the case of the BIRD dataset, our method using GPT-4 achieved a test set execution accuracy of 55.9%, setting a new SOTA.

### 5.5.2 Development set results

Most of our evaluation during development was conducted on the development set of Spider which was easily accessible unlike the test set that was only accessible through an evaluation server provided by Yu et al. [2018]. Table 4a shows the performance of our method using different LLMs, compared to zero-shot prompting of Rajkumar et al. [2022] and Liu et al. [2023a] and our own few-shot prompting. To ensure a fair comparison for the few-shot prompting, we incorporate all the examples utilized for our three classes (easy, non-nested complex, and nested complex) inside the prompt. Given that the CodeX Cushman model has a smaller input context size than the CodeX Davinci and the GPT-4 models, we only use 2 examples from each class (for a total of 6 examples).

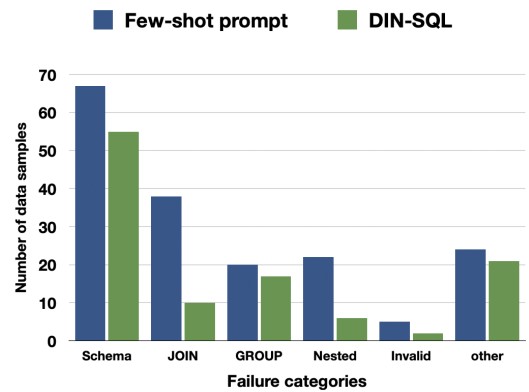

Figure 4: The break-down of failure cases for DIN-SQL (green) and the basic few-shot prompting (blue) across different categories

Our method significantly outperforms both simple few-shot prompting and zero-shot prompting, in terms of both exact set match and execution accuracies, and the improvement is consistent across all models despite their sizes. For example, compared to few-shot prompting, our method improves the execution accuracy for all models by at least 10%.

On the development set of BIRD, our approach demonstrates a substantial improvement, achieving a 4% gain in execution accuracy and a remarkable 9% improvement in valid efficiency score over a GPT-4 baseline Li et al. [2023c], establishing a new SOTA. These and other results are reported in able 3b.

The performance of our method on the test set (as reported in Tables 2 and 3a) is higher than that on the development set for both Spider and BIRD. It is hard to pinpoint the exact reason when the test set is hidden, but we speculate that fewer questions in the test set may require the knowledge of the database content, making it easier for our method to predict a correct SQL query. Furthermore, the development set has schema ambiguity (e.g., a query entity can be mapped to multiple database entities but only one is considered correct), and it is possible that the test set has less ambiguity.

We further analyzed the performance of our proposed method on queries with different levels of difficulty. Table 4b presents the performance of our proposed method compared to a basic few-shot prompting on the development set of Spider. Our proposed method outperforms the basic few-shot prompting across all difficulty levels, with the greatest improvement in performance observed for the extra hard and hard classes where the few-shot prompting performed poorly. Our improvement

| Model | VES | EX |
|---|---|---|
| DIN-SQL + GPT-4 (Ours) | 59.44 | **55.9** |
| GPT-4 | **60.77** | 54.89 |
| Claude-2 | - | 49.02 |
| ChatGPT + CoT [Li et al., 2023c] | 56.56 | 40.08 |
| ChatGPT | 51.40 | 39.30 |
| Codex | 41.60 | 36.47 |
| Palm-2 | - | 33.04 |
| T5-3B | 27.80 | 24.05 |
| T5-Large | 25 | 20.94 |
| T5-Base | 14.7 | 12.89 |

(a) Execution accuracy (EX) and Valid Efficiency Score (VES) on the holdout test set of BIRD

| Model | VES | EX |
|---|---|---|
| DIN-SQL + GPT-4 (Ours) | **58.79** | **50.72** |
| GPT-4 | 49.77 | 46.35 |
| Claude-2 | - | 42.70 |
| ChatGPT + CoT [Li et al., 2023c] | 42.30 | 36.64 |
| ChatGPT | 43.81 | 37.22 |
| Codex | 43.41 | 34.35 |
| Palm-2 | - | 27.38 |
| T5-3B | 25.57 | 23.34 |
| T5-Large | 22.74 | 19.75 |
| T5-Base | 12.90 | 11.54 |

(b) Execution accuracy (EX) and Valid Efficiency Score (VES) on the development set of BIRD

Table 3: Performance of DIN-SQL on BIRD development set and test set.

on the easy class (compared to basic few-shot) is due to incorporating schema links in the prompt, highlighting the importance of our schema-linking module.

| Prompting | Model | EX | EM |
|---|---|---|---|
| DIN-SQL (Ours) | GPT-4 | **74.2** | **60.1** |
| | CodeX Davinci | 69.9 | 57.2 |
| | CodeX Cushman | 47.6 | 35.7 |
| Few-shot (Ours) | GPT-4 | 67.4 | 54.3 |
| | CodeX Davinci | 61.5 | 50.2 |
| | CodeX Cushman | 43.1 | 30.9 |
| Zero-shot (Ours) | GPT-4 | 64.9 | 40.4 |
| Zero-shot [Liu et al., 2023a] | ChatGPT | 60.1 | - |
| Zero-shot [Rajkumar et al., 2022] | CodeX Davinci | 47.5 | |
| Zero-shot (DB content used) [Rajkumar et al., 2022] | CodeX Davinci | 55.1 | |
| | CodeX Cushman | 53 | |
| | GPT3 | 21.7 | |

(a) Performance compared to zero-shot and few-shot prompting using different LLMs on the dev set of Spider

| | Execution accuracy (EX) | | | | | |
|---|---|---|---|---|---|---|
| Prompting | Model | Easy | Medium | Hard | Extra | All |
| DIN-SQL | GPT-4 | **91.1** | **79.8** | **64.9** | **43.4** | **74.2** |
| DIN-SQL | CodeX Davinci | 89.1 | 75.6 | 58 | 38.6 | 69.9 |
| Few-shot | GPT-4 | 86.7 | 73.1 | 59.2 | 31.9 | 67.4 |
| Few-shot | CodeX Davinci | 84.7 | 67.3 | 47.1 | 26.5 | 61.5 |
| | Exact set match accuracy (EM) | | | | | |
| Prompting | Model | Easy | Medium | Hard | Extra | All |
| DIN-SQL | GPT-4 | 82.7 | 65.5 | 42 | **30.7** | 60.1 |
| DIN-SQL | CodeX Davinci | 78.6 | **67.3** | 38.5 | 17.5 | 57.2 |
| Few-shot | GPT-4 | **87.9** | 54 | **47.1** | 12 | 54.3 |
| Few-shot | CodeX Davinci | 77 | 53.8 | 38.5 | 12.7 | 50.2 |

(b) Performance compared to our basic few-shot prompting across different query difficulty levels

Table 4: Performance of DIN-SQL against other in-context learning approaches

### 5.5.3 Error improvements

In Section 3, we did an error analysis of basic few-shot prompting on 500 queries randomly chosen from the training set. To understand the degree those errors are resolved, we ran DIN-SQL on the same 500 queries. As shown in Figure 4, our proposed approach improves the performance for all categories with the largest improvement seen for the JOIN and Nested categories. Despite having an explicit module for schema-linking, the largest portion of failure cases still belong to this category.

### 5.6 Ablation study

In an ablation study, we evaluated our approach with and without each of the four modules. As shown in Table 5 for the CodeX Davinci model, excluding any of the modules leads to an overall decrease in performance, in terms of the execution accuracy.

More details emerge as we study the effectiveness of each module across different query classes. Schema linking helps all query classes with the least improvement for the hard class. Our inspection of a sample of the failed cases reveals that schema linking sometimes finds redundant links due to an ambiguity in the question or schema, and this can introduce redundant joins or output columns.

Without a classification, we had to use either a simple few-shot prompting or a decomposed chain-of-thought (COT) prompting for all queries. The reported performance without a classification

| Prompting | Model | Easy | Medium | Hard | Extra | All |
|---|---|---|---|---|---|---|
| DIN-SQL (generic self-corr) | CodeX Davinci | **89.1** | 75.6 | **58** | **38.6** | 69.9 |
| DIN-SQL (gentle self-corr) | CodeX Davinci | 87.5 | **76.9** | 51.7 | 36.1 | 68.7 |
| DIN-SQL w/o self-corr | CodeX Davinci | 83.9 | 75.4 | 52.3 | 36.1 | 67.3 |
| DIN-SQL w/o schema-linking | CodeX Davinci | 87.3 | 70.6 | 57.6 | 27.1 | 65.9 |
| DIN-SQL w/o classification (simple few-shot prompting) | CodeX Davinci | 87.9 | 68.2 | 51.7 | 27.1 | 63.1 |
| DIN-SQL w/o classification (decomposed COT prompting) | CodeX Davinci | 84.2 | 71.2 | 54.3 | 38.6 | 68.2 |
| DIN-SQL (gentle self-corr) | GPT-4 | **91.1** | **79.8** | **64.9** | **43.4** | **74.2** |
| DIN-SQL (generic self-corr) | GPT-4 | 89.9 | 76.5 | 59.2 | 34.3 | 70.0 |
| DIN-SQL w/o self-correc | GPT-4 | **91.1** | 79.1 | 63.2 | 41.6 | 73.3 |

Table 5: Performance of our method, in terms of execution accuracy, on the dev set with and without each module

module in Table 5 is for our comprehensive framework that includes all our components except classification. This means that the approach contains not only COT prompting but also Schema Linking, Self-Correction, and NatSQL Intermediate Representation, all of which are significant contributions of our work. The decomposed chain-of-thought result presented in this table refers to employing the most complex prompt, developed for the nested complex class, for all questions instead of adopting a classification-based approach to determine prompt complexity based on the question's level of difficulty. In contrast, the result for the DIN-SQL with simple few-shot prompting refers to using the simplest prompting class, easy class, for all questions across different level's of difficulty. As expected, a decomposed chain-of-thought prompting works better for hard and extra hard queries whereas a simple few-shot works better for the easy class.

For self-correction, we ran our study using both CodeX Davinci and GPT-4. For CodeX Davinci, a generic self-correction prompt helps the model across all query classes. A gentle self-correction prompt is also helpful but the gain is smaller than generic one for CodeX Davinci. However, there is less chance that GPT-4 generates a buggy code, and giving a generic prompt of "Buggy SQL:... Fixed SQL:..." can hurt the performance. A gentle prompt work better for GPT-4 and improves the perfromance across all of the classes except the easy class.

# 6   Conclusions

Prompting has enabled large language models to achieve impressive performance on numerous NLP tasks across different domains, without requiring a large training set. Prior to our research, the effectiveness of prompting methods utilizing LLMs for the text-to-SQL task was inferior to that of models fine-tuned for the task. To bridge this gap, we have devised a decomposition technique to tackle some of the challenges that caused this disparity. Our extensive experiments on two challenging datasets of Spider and BIRD show that our method significantly improves the performance of prompting across all query classes, producing comparable or even superior results to state-of-the-art fine-tuned approaches.

# 7   Limitations

There are some limitations or areas of improvement to this work. Our manually constructed demonstrations are fixed for each query class. Future research may explore adaptive and automated methods for generating demonstrations at finer granularities, which can further enhance the performance of our approach. Additionally, as of the time of writing this paper, our proposed approach, characterized by its decomposed and step-by-step structure, incurs a cost of approximately $0.5 and exhibits a latency of approximately 60 seconds when responding to a natural language question from the Spider dataset using GPT-4. We anticipate that as LLMs continue to advance, these costs and latencies should decrease, but reducing the cost is another possible direction.

