## Acknowledgement

This research was funded by Natural Sciences and Engineering Research Council of Canada. We wish to thank Tao Yu and Hongjin Su for running our code on the hold out test set of Spider and Jinyang Li, Binyuan Hui, Reynold Cheng, Ge Qu and the other authors of BIRD for running our code on the holdout test set of BIRD. We also wish to thank Csaba Czepesvari, Dale Schuurmans and the anonymous reviewers of NeurIPS for their constructive comments to improve this work.

## References

Tom Brown, Benjamin Mann, Nick Ryder, Melanie Subbiah, Jared D Kaplan, Prafulla Dhariwal, Arvind Neelakantan, Pranav Shyam, Girish Sastry, Amanda Askell, et al. Language models are few-shot learners. *Advances in neural information processing systems*, 33:1877–1901, 2020.

Ruichu Cai, Jinjie Yuan, Boyan Xu, and Zhifeng Hao. Sadga: Structure-aware dual graph aggregation network for text-to-sql. *Advances in Neural Information Processing Systems*, 34:7664–7676, 2021.

Ruisheng Cao, Lu Chen, Zhi Chen, Yanbin Zhao, Su Zhu, and Kai Yu. Lgesql: line graph enhanced text-to-sql model with mixed local and non-local relations. *arXiv preprint arXiv:2106.01093*, 2021.

Mark Chen, Jerry Tworek, Heewoo Jun, Qiming Yuan, Henrique Ponde de Oliveira Pinto, Jared Kaplan, Harri Edwards, Yuri Burda, Nicholas Joseph, Greg Brockman, et al. Evaluating large language models trained on code. *arXiv preprint arXiv:2107.03374*, 2021.

Wenhu Chen. Large language models are few (1)-shot table reasoners. *arXiv preprint arXiv:2210.06710*, 2022.

DongHyun Choi, Myeong Cheol Shin, EungGyun Kim, and Dong Ryeol Shin. Ryansql: Recursively applying sketch-based slot fillings for complex text-to-sql in cross-domain databases. *Computational Linguistics*, 47(2):309–332, 2021.

Karl Cobbe, Vineet Kosaraju, Mohammad Bavarian, Mark Chen, Heewoo Jun, Lukasz Kaiser, Matthias Plappert, Jerry Tworek, Jacob Hilton, Reiichiro Nakano, et al. Training verifiers to solve math word problems. *arXiv preprint arXiv:2110.14168*, 2021.

Jacob Devlin, Ming-Wei Chang, Kenton Lee, and Kristina Toutanova. Bert: Pre-training of deep bidirectional transformers for language understanding. *arXiv preprint arXiv:1810.04805*, 2018.

Li Dong and Mirella Lapata. Language to logical form with neural attention. In *Proceedings of the 54th Annual Meeting of the Association for Computational Linguistics (Volume 1: Long Papers)*, pages 33–43, 2016.

Yujian Gan, Xinyun Chen, Jinxia Xie, Matthew Purver, John R Woodward, John Drake, and Qiaofu Zhang. Natural sql: Making sql easier to infer from natural language specifications. *arXiv preprint arXiv:2109.05153*, 2021.

Alex Graves and Alex Graves. Long short-term memory. *Supervised sequence labelling with recurrent neural networks*, pages 37–45, 2012.

Jiaqi Guo, Zecheng Zhan, Yan Gao, Yan Xiao, Jian-Guang Lou, Ting Liu, and Dongmei Zhang. Towards complex text-to-sql in cross-domain database with intermediate representation. *arXiv preprint arXiv:1905.08205*, 2019.

Zhixin Guo, Minyxuan Yan, Jiexing Qi, Jianping Zhou, Ziwei He, Zhouhan Lin, Guanjie Zheng, and Xinbing Wang. Few-shot table-to-text generation with prompt planning and knowledge memorization. *arXiv preprint arXiv:2302.04415*, 2023.

Jonathan Herzig, Paweł Krzysztof Nowak, Thomas Müller, Francesco Piccinno, and Julian Martin Eisenschlos. Tapas: Weakly supervised table parsing via pre-training. *arXiv preprint arXiv:2004.02349*, 2020.

Junyang Huang, Yongbo Wang, Yongliang Wang, Yang Dong, and Yanghua Xiao. Relation aware semi-autoregressive semantic parsing for nl2sql. *arXiv preprint arXiv:2108.00804*, 2021.

Binyuan Hui, Xiang Shi, Ruiying Geng, Binhua Li, Yongbin Li, Jian Sun, and Xiaodan Zhu. Improving text-to-sql with schema dependency learning. *arXiv preprint arXiv:2103.04399*, 2021.

Wonseok Hwang, Jinyeong Yim, Seunghyun Park, and Minjoon Seo. A comprehensive exploration on wikisql with table-aware word contextualization. *arXiv preprint arXiv:1902.01069*, 2019.

Rohit Kate. Transforming meaning representation grammars to improve semantic parsing. In *CoNLL 2008: Proceedings of the Twelfth Conference on Computational Natural Language Learning*, pages 33–40, 2008.

Tushar Khot, Harsh Trivedi, Matthew Finlayson, Yao Fu, Kyle Richardson, Peter Clark, and Ashish Sabharwal. Decomposed prompting: A modular approach for solving complex tasks. *arXiv preprint arXiv:2210.02406*, 2022.

Takeshi Kojima, Shixiang Shane Gu, Machel Reid, Yutaka Matsuo, and Yusuke Iwasawa. Large language models are zero-shot reasoners. *arXiv preprint arXiv:2205.11916*, 2022.

Brenden Lake and Marco Baroni. Generalization without systematicity: On the compositional skills of sequence-to-sequence recurrent networks. In *International conference on machine learning*, pages 2873–2882. PMLR, 2018.

Wenqiang Lei, Weixin Wang, Zhixin Ma, Tian Gan, Wei Lu, Min-Yen Kan, and Tat-Seng Chua. Re-examining the role of schema linking in text-to-sql. In *Proceedings of the 2020 Conference on Empirical Methods in Natural Language Processing (EMNLP)*, pages 6943–6954, 2020.

Fei Li and Hosagrahar V Jagadish. Constructing an interactive natural language interface for relational databases. *Proceedings of the VLDB Endowment*, 8(1):73–84, 2014.

Haoyang Li, Jing Zhang, Cuiping Li, and Hong Chen. Decoupling the skeleton parsing and schema linking for text-to-sql. *arXiv preprint arXiv:2302.05965*, 2023a.

Jinyang Li, Binyuan Hui, Reynold Cheng, Bowen Qin, Chenhao Ma, Nan Huo, Fei Huang, Wenyu Du, Luo Si, and Yongbin Li. Graphix-t5: Mixing pre-trained transformers with graph-aware layers for text-to-sql parsing. *arXiv preprint arXiv:2301.07507*, 2023b.

Jinyang Li, Binyuan Hui, Ge Qu, Binhua Li, Jiaxi Yang, Bowen Li, Bailin Wang, Bowen Qin, Rongyu Cao, Ruiying Geng, Nan Huo, Chenhao Ma, Kevin C. C. Chang, Fei Huang, Reynold Cheng, and Yongbin Li. Can llm already serve as a database interface? a big bench for large-scale database grounded text-to-sqls, 2023c.

Yunyao Li, Huahai Yang, and HV Jagadish. Nalix: A generic natural language search environment for xml data. *ACM Transactions on database systems (TODS)*, 32(4):30–es, 2007.

Aiwei Liu, Xuming Hu, Lijie Wen, and Philip S Yu. A comprehensive evaluation of chatgpt's zero-shot text-to-sql capability. *arXiv preprint arXiv:2303.13547*, 2023a.

Pengfei Liu, Weizhe Yuan, Jinlan Fu, Zhengbao Jiang, Hiroaki Hayashi, and Graham Neubig. Pre-train, prompt, and predict: A systematic survey of prompting methods in natural language processing. *ACM Computing Surveys*, 55(9):1–35, 2023b.

Ana-Maria Popescu, Oren Etzioni, and Henry Kautz. Towards a theory of natural language interfaces to databases. In *Proceedings of the 8th international conference on Intelligent user interfaces*, pages 149–157, 2003.

Ana-Maria Popescu, Alex Armanasu, Oren Etzioni, David Ko, and Alexander Yates. Modern natural language interfaces to databases: Composing statistical parsing with semantic tractability. In *COLING 2004: Proceedings of the 20th International Conference on Computational Linguistics*, pages 141–147, 2004.

Jiexing Qi, Jingyao Tang, Ziwei He, Xiangpeng Wan, Chenghu Zhou, Xinbing Wang, Quanshi Zhang, and Zhouhan Lin. Rasat: Integrating relational structures into pretrained seq2seq model for text-to-sql. *arXiv preprint arXiv:2205.06983*, 2022.

Bowen Qin, Binyuan Hui, Lihan Wang, Min Yang, Jinyang Li, Binhua Li, Ruiying Geng, Rongyu Cao, Jian Sun, Luo Si, et al. A survey on text-to-sql parsing: Concepts, methods, and future directions. *arXiv preprint arXiv:2208.13629*, 2022.

Nitarshan Rajkumar, Raymond Li, and Dzmitry Bahdanau. Evaluating the text-to-sql capabilities of large language models. *arXiv preprint arXiv:2204.00498*, 2022.

Ohad Rubin and Jonathan Berant. Smbop: Semi-autoregressive bottom-up semantic parsing. *arXiv preprint arXiv:2010.12412*, 2020.

Torsten Scholak, Nathan Schucher, and Dzmitry Bahdanau. Picard: Parsing incrementally for constrained auto-regressive decoding from language models. *arXiv preprint arXiv:2109.05093*, 2021.

Niculae Stratica, Leila Kosseim, and Bipin C Desai. Using semantic templates for a natural language interface to the cindi virtual library. *Data & Knowledge Engineering*, 55(1):4–19, 2005.

Ilya Sutskever, Oriol Vinyals, and Quoc V Le. Sequence to sequence learning with neural networks. *Advances in neural information processing systems*, 27, 2014.

Bailin Wang, Richard Shin, Xiaodong Liu, Oleksandr Polozov, and Matthew Richardson. Rat-sql: Relation-aware schema encoding and linking for text-to-sql parsers. *arXiv preprint arXiv:1911.04942*, 2019.

Jason Wei, Maarten Bosma, Vincent Y Zhao, Kelvin Guu, Adams Wei Yu, Brian Lester, Nan Du, Andrew M Dai, and Quoc V Le. Finetuned language models are zero-shot learners. *arXiv preprint arXiv:2109.01652*, 2021.

Jason Wei, Yi Tay, Rishi Bommasani, Colin Raffel, Barret Zoph, Sebastian Borgeaud, Dani Yogatama, Maarten Bosma, Denny Zhou, Donald Metzler, et al. Emergent abilities of large language models. *arXiv preprint arXiv:2206.07682*, 2022a.

Jason Wei, Xuezhi Wang, Dale Schuurmans, Maarten Bosma, Ed Chi, Quoc Le, and Denny Zhou. Chain of thought prompting elicits reasoning in large language models. *arXiv preprint arXiv:2201.11903*, 2022b.

Xiaojun Xu, Chang Liu, and Dawn Song. Sqlnet: Generating structured queries from natural language without reinforcement learning. *arXiv preprint arXiv:1711.04436*, 2017.

Kuan Xuan, Yongbo Wang, Yongliang Wang, Zujie Wen, and Yang Dong. Sead: end-to-end text-to-sql generation with schema-aware denoising. *arXiv preprint arXiv:2105.07911*, 2021.

Pengcheng Yin, Graham Neubig, Wen-tau Yih, and Sebastian Riedel. Tabert: Pretraining for joint understanding of textual and tabular data. *arXiv preprint arXiv:2005.08314*, 2020.

Tao Yu, Rui Zhang, Kai Yang, Michihiro Yasunaga, Dongxu Wang, Zifan Li, James Ma, Irene Li, Qingning Yao, Shanelle Roman, et al. Spider: A large-scale human-labeled dataset for complex and cross-domain semantic parsing and text-to-sql task. *arXiv preprint arXiv:1809.08887*, 2018.

Tao Yu, Chien-Sheng Wu, Xi Victoria Lin, Bailin Wang, Yi Chern Tan, Xinyi Yang, Dragomir Radev, Richard Socher, and Caiming Xiong. Grappa: Grammar-augmented pre-training for table semantic parsing. *arXiv preprint arXiv:2009.13845*, 2020.

Lu Zeng, Sree Hari Krishnan Parthasarathi, and Dilek Hakkani-Tur. N-best hypotheses reranking for text-to-sql systems. *arXiv preprint arXiv:2210.10668*, 2022.

Zhuosheng Zhang, Aston Zhang, Mu Li, and Alex Smola. Automatic chain of thought prompting in large language models. *arXiv preprint arXiv:2210.03493*, 2022.

Yiyun Zhao, Jiarong Jiang, Yiqun Hu, Wuwei Lan, Henry Zhu, Anuj Chauhan, Alexander Li, Lin Pan, Jun Wang, Chung-Wei Hang, et al. Importance of synthesizing high-quality data for text-to-sql parsing. *arXiv preprint arXiv:2212.08785*, 2022.

Ruiqi Zhong, Tao Yu, and Dan Klein. Semantic evaluation for text-to-sql with distilled test suite. In *The 2020 Conference on Empirical Methods in Natural Language Processing*. Association for Computational Linguistics, 2020.

Victor Zhong, Caiming Xiong, and Richard Socher. Seq2sql: Generating structured queries from natural language using reinforcement learning. *arXiv preprint arXiv:1709.00103*, 2017.

Denny Zhou, Nathanael Schärli, Le Hou, Jason Wei, Nathan Scales, Xuezhi Wang, Dale Schuurmans, Olivier Bousquet, Quoc Le, and Ed Chi. Least-to-most prompting enables complex reasoning in large language models. *arXiv preprint arXiv:2205.10625*, 2022.

# A Prompts

This section presents a comprehensive list of all the prompts utilized in the four modules of our proposed methodology on both the GPT-4 and CodeX models. The prompts used for each module are provided in detail to allow for easy replication and understanding of the approach. Additionally, we have also included the prompt we used for the few-shot and zero-shot implementations of our method.

For our few-shot examples used for the Non-Nested Complex and Nested Complex classes of queries, we used the NatSQL intermediate representations from the NatSQL Github repository [2]. The repository gives the intermediate representation for all queries in the training set of Spider.

---

[2]https://github.com/ygan/NatSQL

## A.1 Zero-shot prompting

The prompt utilized for the zero-shot prompting scenario draws its inspiration from the work of Liu et al. [2023a], proposed for the ChatGPT. In figure 5, we demonstrate one example for the Zero-shot prompting used in our work.

```
### Complete SQLite SQL QUERY only and with no explanation
### SQLite SQL tables, with their properties:
#
# concert(*, concert_ID, concert_Name, Theme, Stadium_ID, Year)
# singer(*, Singer_ID, Name, Country, Song_Name, Song_release_Year, Age, Is_Male)
# singer_in_concert(*, Concert_ID, Singer_ID)
# stadium(*, Stadium_ID, Location, Name, Capacity, Highest, Lowest, Average)
#
### How many singers do we have?
```

```
SELECT COUNT(*) FROM singer
```

Figure 5: An example of Zero-shot prompting.

## A.2 Few-shot prompting

# Create SQL queries for the given questions.

Table advisor, columns = [*,s_ID,i_ID]
Table classroom, columns = [*,building,room_number,capacity]
Table course, columns = [*,course_id,title,dept_name,credits]
Table department, columns = [*,dept_name,building,budget]
Table instructor, columns = [*,ID,name,dept_name,salary]
Table prereq, columns = [*,course_id,prereq_id]
Table section, columns = [*,course_id,sec_id,semester,year,building,room_number,time_slot_id]
Table student, columns = [*,ID,name,dept_name,tot_cred]
Table takes, columns = [*,ID,course_id,sec_id,semester,year,grade]
Table teaches, columns = [*,ID,course_id,sec_id,semester,year]
Table time_slot, columns = [*,time_slot_id,day,start_hr,start_min,end_hr,end_min]

Q: "Find the buildings which have rooms with capacity more than 50."
SQL: SELECT DISTINCT building FROM classroom WHERE capacity > 50

Q: "Find the room number of the rooms which can sit 50 to 100 students and their buildings."
SQL: SELECT building , room_number FROM classroom WHERE capacity BETWEEN 50 AND 100

Q: "Give the name of the student in the History department with the most credits."
SQL: SELECT name FROM student WHERE dept_name = 'History' ORDER BY tot_cred DESC LIMIT 1

Q: "Find the total budgets of the Marketing or Finance department."
SQL: SELECT sum(budget) FROM department WHERE dept_name = 'Marketing' OR dept_name =

'Finance'

Q: "Find the department name of the instructor whose name contains 'Soisalon'."
SQL: SELECT dept_name FROM instructor WHERE name LIKE '%Soisalon%'

Q: "What is the name of the department with the most credits?"
SQL: SELECT dept_name FROM course GROUP BY dept_name ORDER BY sum(credits) DESC LIMIT 1

Q: "How many instructors teach a course in the Spring of 2010?"
SQL: SELECT COUNT (DISTINCT ID) FROM teaches WHERE semester = 'Spring' AND YEAR = 2010

Q: "Find the name of the students and their department names sorted by their total credits in ascending order."
SQL: SELECT name , dept_name FROM student ORDER BY tot_cred

Q: "Find the year which offers the largest number of courses."
SQL: SELECT YEAR FROM SECTION GROUP BY YEAR ORDER BY count(*) DESC LIMIT 1

Q: "What are the names and average salaries for departments with average salary higher than 42000?"
SQL: SELECT dept_name , AVG (salary) FROM instructor GROUP BY dept_name HAVING AVG (salary) > 42000

Q: "How many rooms in each building have a capacity of over 50?"
SQL: SELECT count(*) , building FROM classroom WHERE capacity > 50 GROUP BY building

Q: "Find the names of the top 3 departments that provide the largest amount of courses?"
SQL: SELECT dept_name FROM course GROUP BY dept_name ORDER BY count(*) DESC LIMIT 3

Q: "Find the maximum and average capacity among rooms in each building."
SQL: SELECT max(capacity) , avg(capacity) , building FROM classroom GROUP BY building

Q: "Find the title of the course that is offered by more than one department."
SQL: SELECT title FROM course GROUP BY title HAVING count(*) > 1

Q: "Find the total budgets of the Marketing or Finance department."
SQL: SELECT sum(budget) FROM department WHERE dept_name = 'Marketing' OR dept_name = 'Finance'

Q: "Find the name and building of the department with the highest budget."
SQL: SELECT dept_name , building FROM department ORDER BY budget DESC LIMIT 1

Q: "What is the name and building of the departments whose budget is more than the average budget?"
SQL: SELECT dept_name , building FROM department WHERE budget > (SELECT avg(budget) FROM department)

Q: "Find the total number of students and total number of instructors for each department."
SQL: SELECT count(DISTINCT T2.id) , count(DISTINCT T3.id) , T3.dept_name FROM department AS T1 JOIN student AS T2 ON T1.dept_name = T2.dept_name JOIN instructor AS T3 ON T1.dept_name = T3.dept_name GROUP BY T3.dept_name

Q: "Find the title of courses that have two prerequisites?"
SQL: SELECT T1.title FROM course AS T1 JOIN prereq AS T2 ON T1.course_id = T2.course_id GROUP BY T2.course_id HAVING count(*) = 2

**Q:** "Find the name of students who took any class in the years of 2009 and 2010."
**SQL:** SELECT DISTINCT T1.name FROM student AS T1 JOIN takes AS T2 ON T1.id = T2.id WHERE T2.YEAR = 2009 OR T2.YEAR = 2010

**Q:** "list in alphabetic order all course names and their instructors' names in year 2008."
**SQL:** SELECT T1.title , T3.name FROM course AS T1 JOIN teaches AS T2 ON T1.course_id = T2.course_id JOIN instructor AS T3 ON T2.id = T3.id WHERE T2.YEAR = 2008 ORDER BY T1.title

**Q:** "Find the title of courses that have two prerequisites?"
**SQL:** SELECT T1.title FROM course AS T1 JOIN prereq AS T2 ON T1.course_id = T2.course_id GROUP BY T2.course_id HAVING count(*) = 2

**Q:** "Find the name and building of the department with the highest budget."
**SQL:** SELECT dept_name , building FROM department ORDER BY budget DESC LIMIT 1

**Q:** "Find the title, credit, and department name of courses that have more than one prerequisites?"
**SQL:** SELECT T1.title , T1.credits , T1.dept_name FROM course AS T1 JOIN prereq AS T2 ON T1.course_id = T2.course_id GROUP BY T2.course_id HAVING count(*) > 1

**Q:** "Give the name and building of the departments with greater than average budget."
**SQL:** SELECT dept_name , building FROM department WHERE budget > (SELECT avg(budget) FROM department)

**Q:** "Find the id of instructors who taught a class in Fall 2009 but not in Spring 2010."
**SQL:** SELECT id FROM teaches WHERE semester = 'Fall' AND YEAR = 2009 EXCEPT SELECT id FROM teaches WHERE semester = 'Spring' AND YEAR = 2010

**Q:** "Find the name of the courses that do not have any prerequisite?"
**SQL:** SELECT title FROM course WHERE course_id NOT IN (SELECT course_id FROM prereq)

**Q:** "Find the salaries of all distinct instructors that are less than the largest salary."
**SQL:** SELECT DISTINCT salary FROM instructor WHERE salary < (SELECT max(salary) FROM instructor)

**Q:** "Find the names of students who have taken any course in the fall semester of year 2003."
**SQL:** SELECT name FROM student WHERE id IN (SELECT id FROM takes WHERE semester = 'Fall' AND YEAR = 2003)

**Q:** "Find the minimum salary for the departments whose average salary is above the average payment of all instructors."
**SQL:** SELECT min(salary) , dept_name FROM instructor GROUP BY dept_name HAVING avg(salary) > (SELECT avg(salary) FROM instructor)

**Q:** "What is the course title of the prerequisite of course Mobile Computing?"
**SQL:** SELECT title FROM course WHERE course_id IN (SELECT T1.prereq_id FROM prereq AS T1 JOIN course AS T2 ON T1.course_id = T2.course_id WHERE T2.title = 'Mobile Computing')

**Q:** "Give the title and credits for the course that is taught in the classroom with the greatest capacity."
**SQL:** SELECT T3.title , T3.credits FROM classroom AS T1 JOIN SECTION AS T2 ON T1.building = T2.building AND T1.room_number = T2.room_number JOIN course AS T3 ON T2.course_id = T3.course_id WHERE T1.capacity = (SELECT max(capacity) FROM classroom)

## A.3 Schema linking prompt

# Find the schema_links for generating SQL queries for each question based on the database schema and Foreign keys.

Table city, columns = [*,City_ID,Official_Name,Status,Area_km_2,Population,Census_Ranking]
Table competition_record, columns = [*,Competition_ID,Farm_ID,Rank]
Table farm, columns = [*,Farm_ID,Year,Total_Horses,Working_Horses,
Total_Cattle,Oxen,Bulls,Cows,Pigs,Sheep_and_Goats]
Table farm_competition, columns = [*,Competition_ID,Year,Theme,Host_city_ID,Hosts]
Foreign_keys = [farm_competition.Host_city_ID = city.City_ID,competition_record.Farm_ID = farm.Farm_ID,competition_record.Competition_ID = farm_competition.Competition_ID]
Q: "Show the status of the city that has hosted the greatest number of competitions."
A: Let's think step by step. In the question "Show the status of the city that has hosted the greatest number of competitions.", we are asked:
"the status of the city" so we need column = [city.Status]
"greatest number of competitions" so we need column = [farm_competition.*]
Based on the columns and tables, we need these Foreign_keys = [farm_competition.Host_city_ID = city.City_ID].
Based on the tables, columns, and Foreign_keys, The set of possible cell values are = []. So the Schema_links are:
Schema_links: [city.Status,farm_competition.Host_city_ID = city.City_ID,farm_competition.*]

Table department, columns = [*,Department_ID,Name,Creation,Ranking,Budget_in_Billions,Num_Employees]
Table head, columns = [*,head_ID,name,born_state,age]
Table management, columns = [*,department_ID,head_ID,temporary_acting]
Foreign_keys = [management.head_ID = head.head_ID,management.department_ID = department.Department_ID]
Q: "How many heads of the departments are older than 56 ?"
A: Let's think step by step. In the question "How many heads of the departments are older than 56 ?", we are asked:
"How many heads of the departments" so we need column = [head.*]
"older" so we need column = [head.age]
Based on the columns and tables, we need these Foreign_keys = [].
Based on the tables, columns, and Foreign_keys, The set of possible cell values are = [56]. So the Schema_links are:
Schema_links: [head.*,head.age,56]

Table department, columns = [*,Department_ID,Name,Creation,Ranking,Budget_in_Billions,Num_Employees]
Table head, columns = [*,head_ID,name,born_state,age]
Table management, columns = [*,department_ID,head_ID,temporary_acting]
Foreign_keys = [management.head_ID = head.head_ID,management.department_ID = department.Department_ID]
Q: "what are the distinct creation years of the departments managed by a secretary born in state 'Alabama'?"
A: Let's think step by step. In the question "what are the distinct creation years of the departments managed by a secretary born in state 'Alabama'?", we are asked:
"distinct creation years of the departments" so we need column = [department.Creation]
"departments managed by" so we need column = [management.department_ID]
"born in" so we need column = [head.born_state]
Based on the columns and tables, we need these Foreign_keys = [department.Department_ID = management.department_ID,management.head_ID = head.head_ID].
Based on the tables, columns, and Foreign_keys, The set of possible cell values are = ['Alabama'].
So the Schema_links are:
Schema_links: [department.Creation,department.Department_ID = management.department_ID, head.head_ID = management.head_ID,head.born_state,'Alabama']

Table Addresses, columns = [*,address_id,line_1,line_2,city,zip_postcode,state_province_county,country]
Table Candidate_Assessments, columns = [*,candidate_id,qualification,assessment_date,asessment_outcome_code]

Table Candidates, columns = [*,candidate_id,candidate_details]
Table Courses, columns = [*,course_id,course_name,course_description,other_details]
Table People, columns = [*,person_id,first_name,middle_name,
last_name,cell_mobile_number,email_address,login_name,password]
Table People_Addresses, columns = [*,person_address_id,person_id,address_id,date_from,date_to]
Table Student_Course_Attendance, columns = [*,student_id,course_id,date_of_attendance]
Table Student_Course_Registrations, columns = [*,student_id,course_id,registration_date]
Table Students, columns = [*,student_id,student_details]
Foreign_keys = [Students.student_id = People.person_id,People_Addresses.address_id = Addresses.address_id,People_Addresses.person_id =
People.person_id,Student_Course_Registrations.course_id =
Courses.course_id,Student_Course_Registrations.student_id =
Students.student_id,Student_Course_Attendance.student_id =
Student_Course_Registrations.student_id,Student_Course_Attendance.course_id = Student_Course_Registrations.course_id,Candidates.candidate_id =
People.person_id,Candidate_Assessments.candidate_id = Candidates.candidate_id]
Q: "List the id of students who never attends courses?"
A: Let's think step by step. In the question "List the id of students who never attends courses?", we
are asked:
"id of students" so we need column = [Students.student_id]
"never attends courses" so we need column = [Student_Course_Attendance.student_id]
Based on the columns and tables, we need these Foreign_keys = [Students.student_id = Student_Course_Attendance.student_id].
Based on the tables, columns, and Foreign_keys, The set of possible cell values are = []. So the
Schema_links are:
Schema_links: [Students.student_id = Student_Course_Attendance.student_id]

Table Country, columns = [*,id,name]
Table League, columns = [*,id,country_id,name]
Table Player, columns = [*,id,player_api_id,player_name,player_fifa_api_id,birthday,height,weight]
Table Player_Attributes, columns = [*,id,player_fifa_api_id,player_api_id,date,overall_rating,potential
,preferred_foot,attacking_work_rate,defensive_work_rate,crossing,finishing
,heading_accuracy,short_passing,volleys,dribbling,curve,free_kick_accuracy
,long_passing,ball_control,acceleration,sprint_speed,agility,reactions,balance
,shot_power,jumping,stamina,strength,long_shots,aggression,interceptions
,positioning,vision,penalties,marking,standing_tackle,sliding_tackle,gk_diving
,gk_handling,gk_kicking,gk_positioning,gk_reflexes]
Table Team, columns = [*,id,team_api_id,team_fifa_api_id,team_long_name,team_short_name]
Table Team_Attributes, columns = [*,id,team_fifa_api_id,team_api_id,date,buildUpPlaySpeed
,buildUpPlaySpeedClass,buildUpPlayDribbling,buildUpPlayDribblingClass
,buildUpPlayPassing,buildUpPlayPassingClass,buildUpPlayPositioningClass,chanceCreationPassing
,chanceCreationPassingClass,chanceCreationCrossing,chanceCreationCrossingClass
,chanceCreationShooting,chanceCreationShootingClass,chanceCreationPositioningClass
,defencePressure,defencePressureClass,defenceAggression,defenceAggressionClass
,defenceTeamWidth,defenceTeamWidthClass,defenceDefenderLineClass]
Table sqlite_sequence, columns = [*,name,seq]
Foreign_keys = [Player_Attributes.player_api_id = Player.player_api_id,
Player_Attributes.player_fifa_api_id = Player.player_fifa_api_id,
League.country_id = Country.id,Team_Attributes.team_api_id = Team.team_api_id,
Team_Attributes.team_fifa_api_id = Team.team_fifa_api_id]
Q: "List the names of all left-footed players who have overall rating between 85 and 90."
A: Let's think step by step. In the question "List the names of all left-footed players who have overall
rating between 85 and 90.", we are asked:
"names of all left-footed players" so we need column =
[Player.player_name,Player_Attributes.preferred_foot]
"players who have overall rating" so we need column = [Player_Attributes.overall_rating]
Based on the columns and tables, we need these Foreign_keys = [Player_Attributes.player_api_id =
Player.player_api_id].
Based on the tables, columns, and Foreign_keys, The set of possible cell values are = [left,85,90]. So

the Schema_links are:
Schema_links: [Player.player_name,Player_Attributes.preferred_foot,Player_Attributes.overall_rating,
Player_Attributes.player_api_id = Player.player_api_id,left,85,90]

Table advisor, columns = [*,s_ID,i_ID]
Table classroom, columns = [*,building,room_number,capacity]
Table course, columns = [*,course_id,title,dept_name,credits]
Table department, columns = [*,dept_name,building,budget]
Table instructor, columns = [*,ID,name,dept_name,salary]
Table prereq, columns = [*,course_id,prereq_id]
Table section, columns = [*,course_id,sec_id,semester,year,building,room_number,time_slot_id]
Table student, columns = [*,ID,name,dept_name,tot_cred]
Table takes, columns = [*,ID,course_id,sec_id,semester,year,grade]
Table teaches, columns = [*,ID,course_id,sec_id,semester,year]
Table time_slot, columns = [*,time_slot_id,day,start_hr,start_min,end_hr,end_min]
Foreign_keys = [course.dept_name = department.dept_name,instructor.dept_name = department.dept_name,section.building = classroom.building
,section.room_number = classroom.room_number
,section.course_id = course.course_id,teaches.ID = instructor.ID,teaches.course_id = section.course_id,teaches.sec_id = section.sec_id,
teaches.semester = section.semester,teaches.year = section.year,student.dept_name = department.dept_name,
takes.ID = student.ID,takes.course_id = section.course_id,
takes.sec_id = section.sec_id,takes.semester = section.semester,
takes.year = section.year,advisor.s_ID = student.ID,
advisor.i_ID = instructor.ID,prereq.prereq_id = course.course_id,prereq.course_id = course.course_id]
Q: "Give the title of the course offered in Chandler during the Fall of 2010."
A: Let's think step by step. In the question "Give the title of the course offered in Chandler during the Fall of 2010.", we are asked:
"title of the course" so we need column = [course.title]
"course offered in Chandler" so we need column = [SECTION.building]
"during the Fall" so we need column = [SECTION.semester]
"of 2010" so we need column = [SECTION.year]
Based on the columns and tables, we need these Foreign_keys = [course.course_id = SECTION.course_id].
Based on the tables, columns, and Foreign_keys, The set of possible cell values are = [Chandler,Fall,2010]. So the Schema_links are:
Schema_links: [course.title,course.course_id = SECTION.course_id,SECTION.building,SECTION.year
,SECTION.semester,Chandler,Fall,2010]

Table advisor, columns = [*,s_ID,i_ID]
Table classroom, columns = [*,building,room_number,capacity]
Table course, columns = [*,course_id,title,dept_name,credits]
Table department, columns = [*,dept_name,building,budget]
Table instructor, columns = [*,ID,name,dept_name,salary]
Table prereq, columns = [*,course_id,prereq_id]
Table section, columns = [*,course_id,sec_id,semester,year,building,room_number,time_slot_id]
Table student, columns = [*,ID,name,dept_name,tot_cred]
Table takes, columns = [*,ID,course_id,sec_id,semester,year,grade]
Table teaches, columns = [*,ID,course_id,sec_id,semester,year]
Table time_slot, columns = [*,time_slot_id,day,start_hr,start_min,end_hr,end_min]
Foreign_keys = [course.dept_name = department.dept_name,instructor.dept_name = department.dept_name,
section.building = classroom.building,section.room_number = classroom.room_number,
section.course_id = course.course_id,teaches.ID = instructor.ID,teaches.course_id = section.course_id,
teaches.sec_id = section.sec_id,teaches.semester = section.semester,teaches.year = section.year,
student.dept_name = department.dept_name,takes.ID = student.ID,takes.course_id = sec-

tion.course_id,
takes.sec_id = section.sec_id,takes.semester = section.semester,
takes.year = section.year,advisor.s_ID = student.ID,advisor.i_ID = instructor.ID,
prereq.prereq_id = course.course_id,prereq.course_id = course.course_id]
Q: "Find the id of instructors who taught a class in Fall 2009 but not in Spring 2010."
A: Let's think step by step. In the question "Find the id of instructors who taught a class in Fall 2009 but not in Spring 2010.", we are asked:
"id of instructors who taught " so we need column = [teaches.id]
"taught a class in" so we need column = [teaches.semester,teaches.year]
Based on the columns and tables, we need these Foreign_keys = [].
Based on the tables, columns, and Foreign_keys, The set of possible cell values are = [Fall,2009,Spring,2010]. So the Schema_links are:
Schema_links: [teaches.id,teaches.semester,teaches.year,Fall,2009,Spring,2010]

Table Accounts, columns = [*,account_id,customer_id,date_account_opened,account_name,other_account_details]
Table Customers, columns = [*,customer_id,customer_first_name,customer_middle_initial,
customer_last_name,gender,email_address,login_name,login_password,phone_number,
town_city,state_county_province,country]
Table Financial_Transactions, columns = [*,transaction_id,account_id,invoice_number,transaction_type,
transaction_date,transaction_amount,transaction_comment,other_transaction_details]
Table Invoice_Line_Items, columns = [*,order_item_id,invoice_number,product_id,product_title,product_quantity
,product_price,derived_product_cost,derived_vat_payable,derived_total_cost]
Table Invoices, columns = [*,invoice_number,order_id,invoice_date]
Table Order_Items, columns = [*,order_item_id,order_id,product_id,product_quantity,other_order_item_details]
Table Orders, columns = [*,order_id,customer_id,date_order_placed,order_details]
Table Product_Categories, columns = [*,production_type_code,product_type_description,vat_rating]
Table Products, columns = [*,product_id,parent_product_id,production_type_code
,unit_price,product_name,product_color,product_size]
Foreign_keys = [Orders.customer_id = Customers.customer_id,Invoices.order_id = Or-ders.order_id,Accounts.customer_id = Customers.customer_id,
Products.production_type_code = Product_Categories.production_type_code,Financial_Transactions.account_id = Accounts.account_id,Financial_Transactions.invoice_number = Invoices.invoice_number,
Order_Items.order_id = Orders.order_id,Order_Items.product_id = Products.product_id,
Invoice_Line_Items.product_id = Products.product_id,Invoice_Line_Items.invoice_number = Invoices.invoice_number,
Invoice_Line_Items.order_item_id = Order_Items.order_item_id]
Q: "Show the id, the date of account opened, the account name, and other account detail for all accounts."
A: Let's think step by step. In the question "Show the id, the date of account opened, the account name, and other account detail for all accounts.", we are asked:
"the id, the date of account opened, the account name, and other account detail for all accounts." so we need column = [Accounts.account_id,
Accounts.account_name,Accounts.other_account_details,Accounts.date_account_opened]
Based on the columns and tables, we need these Foreign_keys = [].
Based on the tables, columns, and Foreign_keys, The set of possible cell values are = []. So the Schema_links are:
Schema_links: [Accounts.account_id,Accounts.account_name,
Accounts.other_account_details,Accounts.date_account_opened]

Table advisor, columns = [*,s_ID,i_ID]
Table classroom, columns = [*,building,room_number,capacity]
Table course, columns = [*,course_id,title,dept_name,credits]
Table department, columns = [*,dept_name,building,budget]
Table instructor, columns = [*,ID,name,dept_name,salary]
Table prereq, columns = [*,course_id,prereq_id]
Table section, columns = [*,course_id,sec_id,semester,year,building,room_number,time_slot_id]
Table student, columns = [*,ID,name,dept_name,tot_cred]
Table takes, columns = [*,ID,course_id,sec_id,semester,year,grade]
Table teaches, columns = [*,ID,course_id,sec_id,semester,year]

Table time_slot, columns = [*,time_slot_id,day,start_hr,start_min,end_hr,end_min]
Foreign_keys = [course.dept_name = department.dept_name,instructor.dept_name = depart-ment.dept_name,
section.building = classroom.building,section.room_number = classroom.room_number,
section.course_id = course.course_id,teaches.ID = instructor.ID,teaches.course_id = sec-tion.course_id,teaches.sec_id = section.sec_id,
teaches.semester = section.semester,teaches.year = section.year,student.dept_name = depart-ment.dept_name,takes.ID = student.ID,takes.course_id = section.course_id,
takes.sec_id = section.sec_id,takes.semester = section.semester,takes.year = section.year,advisor.s_ID = student.ID,
advisor.i_ID = instructor.ID,prereq.prereq_id = course.course_id,prereq.course_id = course.course_id]
Q: "Find the buildings which have rooms with capacity more than 50."
A: Let's think step by step. In the question "Find the buildings which have rooms with capacity more than 50.", we are asked:
"the buildings which have rooms" so we need column = [classroom.capacity]
"rooms with capacity" so we need column = [classroom.building]
Based on the columns and tables, we need these Foreign_keys = [].
Based on the tables, columns, and Foreign_keys, The set of possible cell values are = [50]. So the Schema_links are:
Schema_links: [classroom.building,classroom.capacity,50]

Table city, columns = [*,City_ID,Official_Name,Status,Area_km_2,Population,Census_Ranking]
Table competition_record, columns = [*,Competition_ID,Farm_ID,Rank]
Table farm, columns = [*,Farm_ID,Year,Total_Horses,Working_Horses,Total_Cattle,Oxen,Bulls,Cows,Pigs,Sheep_and_Goats]
Table farm_competition, columns = [*,Competition_ID,Year,Theme,Host_city_ID,Hosts]
Foreign_keys = [farm_competition.Host_city_ID = city.City_ID,competition_record.Farm_ID = farm.Farm_ID,competition_record.Competition_ID = farm_competition.Competition_ID]
Q: "Show the status shared by cities with population bigger than 1500 and smaller than 500."
A: Let's think step by step. In the question "Show the status shared by cities with population bigger than 1500 and smaller than 500.", we are asked:
"the status shared by cities" so we need column = [city.Status]
"cities with population" so we need column = [city.Population]
Based on the columns and tables, we need these Foreign_keys = [].
Based on the tables, columns, and Foreign_keys, The set of possible cell values are = [1500,500]. So the Schema_links are:
Schema_links: [city.Status,city.Population,1500,500]

## A.4   Classification & decomposition prompt

# For the given question, classify it as EASY, NON-NESTED, or NESTED based on nested queries and JOIN.

if need nested queries: predict NESTED
elif need JOIN and don't need nested queries: predict NON-NESTED
elif don't need JOIN and don't need nested queries: predict EASY

Table advisor, columns = [*,s_ID,i_ID]
Table classroom, columns = [*,building,room_number,capacity]
Table course, columns = [*,course_id,title,dept_name,credits]
Table department, columns = [*,dept_name,building,budget]
Table instructor, columns = [*,ID,name,dept_name,salary]
Table prereq, columns = [*,course_id,prereq_id]
Table section, columns = [*,course_id,sec_id,semester,year,building,room_number,time_slot_id]
Table student, columns = [*,ID,name,dept_name,tot_cred]
Table takes, columns = [*,ID,course_id,sec_id,semester,year,grade]
Table teaches, columns = [*,ID,course_id,sec_id,semester,year]
Table time_slot, columns = [*,time_slot_id,day,start_hr,start_min,end_hr,end_min]

Foreign_keys = [course.dept_name = department.dept_name,instructor.dept_name = department.dept_name,section.building = classroom.building,section.room_number = classroom.room_number,
section.course_id = course.course_id,teaches.ID = instructor.ID,teaches.course_id = section.course_id,teaches.sec_id = section.sec_id,teaches.semester = section.semester,
teaches.year = section.year,student.dept_name = department.dept_name,takes.ID = student.ID,takes.course_id = section.course_id,takes.sec_id = section.sec_id,takes.semester = section.semester,takes.year = section.year,advisor.s_ID = student.ID,
advisor.i_ID = instructor.ID,prereq.prereq_id = course.course_id,prereq.course_id = course.course_id]

Q: "Find the buildings which have rooms with capacity more than 50."
schema_links: [classroom.building,classroom.capacity,50]
A: Let's think step by step. The SQL query for the question "Find the buildings which have rooms with capacity more than 50." needs these tables = [classroom], so we don't need JOIN.
Plus, it doesn't require nested queries with (INTERSECT, UNION, EXCEPT, IN, NOT IN), and we need the answer to the questions = [""].
So, we don't need JOIN and don't need nested queries, then the the SQL query can be classified as "EASY".
Label: "EASY"

Q: "What are the names of all instructors who advise students in the math depart sorted by total credits of the student."
schema_links: [advisor.i_id = instructor.id,advisor.s_id = student.id,instructor.name, student.dept_name,student.tot_cred,math]
A: Let's think step by step. The SQL query for the question "What are the names of all instructors who advise students in the math depart sorted by total credits of the student." needs these tables = [advisor,instructor,student], so we need JOIN.
Plus, it doesn't need nested queries with (INTERSECT, UNION, EXCEPT, IN, NOT IN), and we need the answer to the questions = [""].
So, we need JOIN and don't need nested queries, then the the SQL query can be classified as "NON-NESTED".
Label: "NON-NESTED"

Q: "How many courses that do not have prerequisite?"
schema_links: [course.*,course.course_id = prereq.course_id]
A: Let's think step by step. The SQL query for the question "How many courses that do not have prerequisite?" needs these tables = [course,prereq], so we need JOIN.
Plus, it requires nested queries with (INTERSECT, UNION, EXCEPT, IN, NOT IN), and we need the answer to the questions = ["Which courses have prerequisite?"].
So, we need JOIN and need nested queries, then the the SQL query can be classified as "NESTED".
Label: "NESTED"

Q: "Find the room number of the rooms which can sit 50 to 100 students and their buildings."
schema_links: [classroom.building,classroom.room_number,classroom.capacity,50,100]
A: Let's think step by step. The SQL query for the question "Find the room number of the rooms which can sit 50 to 100 students and their buildings." needs these tables = [classroom], so we don't need JOIN.
Plus, it doesn't require nested queries with (INTERSECT, UNION, EXCEPT, IN, NOT IN), and we need the answer to the questions = [""].
So, we don't need JOIN and don't need nested queries, then the the SQL query can be classified as "EASY".
Label: "EASY"

Q: "Find the title of course that is provided by both Statistics and Psychology departments."
schema_links: [course.title,course.dept_name,Statistics,Psychology]
A: Let's think step by step. The SQL query for the question "Find the title of course that is provided

by both Statistics and Psychology departments." needs these tables = [course], so we don't need JOIN.

Plus, it requires nested queries with (INTERSECT, UNION, EXCEPT, IN, NOT IN), and we need the answer to the questions = ["Find the titles of courses that is provided by Psychology departments"]. So, we don't need JOIN and need nested queries, then the the SQL query can be classified as "NESTED".

Label: "NESTED"

Q: "Find the id of instructors who taught a class in Fall 2009 but not in Spring 2010."

schema_links: [teaches.id,teaches.semester,teaches.year,Fall,2009,Spring,2010]

A: Let's think step by step. The SQL query for the question "Find the id of instructors who taught a class in Fall 2009 but not in Spring 2010." needs these tables = [teaches], so we don't need JOIN.

Plus, it requires nested queries with (INTERSECT, UNION, EXCEPT, IN, NOT IN), and we need the answer to the questions = ["Find the id of instructors who taught a class in Spring 2010"]. So, we don't need JOIN and need nested queries, then the the SQL query can be classified as "NESTED".

Label: "NESTED"

Q: "Find the name of the department that offers the highest total credits?"

schema_links: [course.dept_name,course.credits]

A: Let's think step by step. The SQL query for the question "Find the name of the department that offers the highest total credits?." needs these tables = [course], so we don't need JOIN.

Plus, it doesn't require nested queries with (INTERSECT, UNION, EXCEPT, IN, NOT IN), and we need the answer to the questions = [""].

So, we don't need JOIN and don't need nested queries, then the the SQL query can be classified as "EASY".

Label: "EASY"

Q: "What is the name of the instructor who advises the student with the greatest number of total credits?"

schema_links: [advisor.i_id = instructor.id,advisor.s_id = student.id,instructor.name,student.tot_cred ]

A: Let's think step by step. The SQL query for the question "What is the name of the instructor who advises the student with the greatest number of total credits?" needs these tables = [advisor,instructor,student], so we need JOIN.

Plus, it doesn't need nested queries with (INTERSECT, UNION, EXCEPT, IN, NOT IN), and we need the answer to the questions = [""].

So, we need JOIN and don't need nested queries, then the the SQL query can be classified as "NON-NESTED".

Label: "NON-NESTED"

Q: "Find the total number of students and total number of instructors for each department."

schema_links: [department.dept_name = instructor.dept_name,student.id,student.dept_name = department.dept_name,instructor.id]

A: Let's think step by step. The SQL query for the question "Find the total number of students and total number of instructors for each department." needs these tables = [department,instructor,student], so we need JOIN.

Plus, it doesn't need nested queries with (INTERSECT, UNION, EXCEPT, IN, NOT IN), and we need the answer to the questions = [""].

So, we need JOIN and don't need nested queries, then the the SQL query can be classified as "NON-NESTED".

Label: "NON-NESTED"

Q: "Give the name and building of the departments with greater than average budget."

schema_links: [department.budget,department.dept_name,department.building]

A: Let's think step by step. The SQL query for the question "Give the name and building of the departments with greater than average budget." needs these tables = [department], so we don't need JOIN.

Plus, it requires nested queries with (INTERSECT, UNION, EXCEPT, IN, NOT IN), and we need

the answer to the questions = ["What is the average budget of the departments"].
So, we don't need JOIN and need nested queries, then the the SQL query can be classified as "NESTED".
Label: "NESTED"

## A.5   SQL generation

### A.5.1   Easy Class

# Use the the schema links to generate the SQL queries for each of the questions.

Table advisor, columns = [*,s_ID,i_ID]
Table classroom, columns = [*,building,room_number,capacity]
Table course, columns = [*,course_id,title,dept_name,credits]
Table department, columns = [*,dept_name,building,budget]
Table instructor, columns = [*,ID,name,dept_name,salary]
Table prereq, columns = [*,course_id,prereq_id]
Table section, columns = [*,course_id,sec_id,semester,year,building,room_number,time_slot_id]
Table student, columns = [*,ID,name,dept_name,tot_cred]
Table takes, columns = [*,ID,course_id,sec_id,semester,year,grade]
Table teaches, columns = [*,ID,course_id,sec_id,semester,year]
Table time_slot, columns = [*,time_slot_id,day,start_hr,start_min,end_hr,end_min]

Q: "Find the buildings which have rooms with capacity more than 50."
Schema_links: [classroom.building,classroom.capacity,50]
SQL: SELECT DISTINCT building FROM classroom WHERE capacity > 50

Q: "Find the room number of the rooms which can sit 50 to 100 students and their buildings."
Schema_links: [classroom.building,classroom.room_number,classroom.capacity,50,100]
SQL: SELECT building , room_number FROM classroom WHERE capacity BETWEEN 50 AND 100

Q: "Give the name of the student in the History department with the most credits."
Schema_links: [student.name,student.dept_name,student.tot_cred,History]
SQL: SELECT name FROM student WHERE dept_name = 'History' ORDER BY tot_cred DESC LIMIT 1

Q: "Find the total budgets of the Marketing or Finance department."
Schema_links: [department.budget,department.dept_name,Marketing,Finance]
SQL: SELECT sum(budget) FROM department WHERE dept_name = 'Marketing' OR dept_name = 'Finance'

Q: "Find the department name of the instructor whose name contains 'Soisalon'."
Schema_links: [instructor.dept_name,instructor.name,Soisalon]
SQL: SELECT dept_name FROM instructor WHERE name LIKE '%Soisalon%'

Q: "What is the name of the department with the most credits?"
Schema_links: [course.dept_name,course.credits]
SQL: SELECT dept_name FROM course GROUP BY dept_name ORDER BY sum(credits) DESC LIMIT 1

Q: "How many instructors teach a course in the Spring of 2010?"
Schema_links: [teaches.ID,teaches.semester,teaches.YEAR,Spring,2010]
SQL: SELECT COUNT (DISTINCT ID) FROM teaches WHERE semester = 'Spring' AND YEAR = 2010

Q: "Find the name of the students and their department names sorted by their total credits in ascending order."
Schema_links: [student.name,student.dept_name,student.tot_cred]
SQL: SELECT name , dept_name FROM student ORDER BY tot_cred

Q: "Find the year which offers the largest number of courses."
Schema_links: [SECTION.YEAR,SECTION.*]
SQL: SELECT YEAR FROM SECTION GROUP BY YEAR ORDER BY count(*) DESC LIMIT 1

Q: "What are the names and average salaries for departments with average salary higher than 42000?"
Schema_links: [instructor.dept_name,instructor.salary,42000]
SQL: SELECT dept_name , AVG (salary) FROM instructor GROUP BY dept_name HAVING AVG (salary) > 42000

Q: "How many rooms in each building have a capacity of over 50?"
Schema_links: [classroom.*,classroom.building,classroom.capacity,50]
SQL: SELECT count(*) , building FROM classroom WHERE capacity > 50 GROUP BY building

Q: "Find the names of the top 3 departments that provide the largest amount of courses?"
Schema_links: [course.dept_name,course.*]
SQL: SELECT dept_name FROM course GROUP BY dept_name ORDER BY count(*) DESC LIMIT 3

Q: "Find the maximum and average capacity among rooms in each building."
Schema_links: [classroom.building,classroom.capacity]
SQL: SELECT max(capacity) , avg(capacity) , building FROM classroom GROUP BY building

Q: "Find the title of the course that is offered by more than one department."
Schema_links: [course.title]
SQL: SELECT title FROM course GROUP BY title HAVING count(*) > 1

## A.5.2   Non-Nested Complex

# Use the the schema links and Intermediate_representation to generate the SQL queries for each of the questions.

Table advisor, columns = [*,s_ID,i_ID]
Table classroom, columns = [*,building,room_number,capacity]
Table course, columns = [*,course_id,title,dept_name,credits]
Table department, columns = [*,dept_name,building,budget]
Table instructor, columns = [*,ID,name,dept_name,salary]
Table prereq, columns = [*,course_id,prereq_id]
Table section, columns = [*,course_id,sec_id,semester,year,building,room_number,time_slot_id]
Table student, columns = [*,ID,name,dept_name,tot_cred]
Table takes, columns = [*,ID,course_id,sec_id,semester,year,grade]
Table teaches, columns = [*,ID,course_id,sec_id,semester,year]
Table time_slot, columns = [*,time_slot_id,day,start_hr,start_min,end_hr,end_min]
Foreign_keys = [course.dept_name = department.dept_name,instructor.dept_name = department.dept_name,section.building = classroom.building,
section.room_number = classroom.room_number,section.course_id = course.course_id,teaches.ID = instructor.ID,teaches.course_id = section.course_id,
teaches.sec_id = section.sec_id,teaches.semester = section.semester,teaches.year = section.year,
student.dept_name = department.dept_name,takes.ID = student.ID,takes.course_id = section.course_id,takes.sec_id = section.sec_id,takes.semester = section.semester,
takes.year = section.year,advisor.s_ID = student.ID,advisor.i_ID = instructor.ID,prereq.prereq_id = course.course_id,

prereq.course_id = course.course_id]

Q: "Find the total budgets of the Marketing or Finance department."
Schema_links: [department.budget,department.dept_name,Marketing,Finance]
A: Let's think step by step. For creating the SQL for the given question, we need to join these tables = [].
First, create an intermediate representation, then use it to construct the SQL query.
Intermediate_representation: select sum(department.budget) from department where department.dept_name = "Marketing" or department.dept_name = "Finance"
SQL: SELECT sum(budget) FROM department WHERE dept_name = 'Marketing' OR dept_name = 'Finance'

Q: "Find the name and building of the department with the highest budget."
Schema_links: [department.budget,department.dept_name,department.building]
A: Let's think step by step. For creating the SQL for the given question, we need to join these tables = [].
First, create an intermediate representation, then use it to construct the SQL query.
Intermediate_representation: select department.dept_name , department.building from department order by department.budget desc limit 1
SQL: SELECT dept_name , building FROM department ORDER BY budget DESC LIMIT 1

Q: "What is the name and building of the departments whose budget is more than the average budget?"
Schema_links: [department.budget,department.dept_name,department.building]
A: Let's think step by step. For creating the SQL for the given question, we need to join these tables = [].
First, create an intermediate representation, then use it to construct the SQL query.
Intermediate_representation: select department.dept_name , department.building from department where @.@ > avg ( department.budget )
SQL: SELECT dept_name , building FROM department WHERE budget > (SELECT avg(budget) FROM department)

Q: "Find the total number of students and total number of instructors for each department."
Schema_links: [department.dept_name = student.dept_name,student.id,department.dept_name = instructor.dept_name,instructor.id]
A: Let's think step by step. For creating the SQL for the given question, we need to join these tables = [department,student,instructor].
First, create an intermediate representation, then use it to construct the SQL query.
Intermediate_representation: "select count( distinct student.ID) , count( distinct instructor.ID) , department.dept_name from department group by instructor.dept_name
SQL: SELECT count(DISTINCT T2.id) , count(DISTINCT T3.id) , T3.dept_name FROM department AS T1 JOIN student AS T2 ON T1.dept_name = T2.dept_name JOIN instructor AS T3 ON T1.dept_name = T3.dept_name GROUP BY T3.dept_name

Q: "Find the title of courses that have two prerequisites?"
Schema_links: [course.title,course.course_id = prereq.course_id]
A: Let's think step by step. For creating the SQL for the given question, we need to join these tables = [course,prereq].
First, create an intermediate representation, then use it to construct the SQL query.
Intermediate_representation: select course.title from course where count ( prereq.* ) = 2 group by prereq.course_id
SQL: SELECT T1.title FROM course AS T1 JOIN prereq AS T2 ON T1.course_id = T2.course_id GROUP BY T2.course_id HAVING count(*) = 2

Q: "Find the name of students who took any class in the years of 2009 and 2010."
Schema_links: [student.name,student.id = takes.id,takes.YEAR,2009,2010]
A: Let's think step by step. For creating the SQL for the given question, we need to join these tables = [student,takes].

First, create an intermediate representation, then use it to construct the SQL query.

Intermediate_representation: select distinct student.name from student where takes.year = 2009 or takes.year = 2010

SQL: SELECT DISTINCT T1.name FROM student AS T1 JOIN takes AS T2 ON T1.id = T2.id WHERE T2.YEAR = 2009 OR T2.YEAR = 2010

Q: "list in alphabetic order all course names and their instructors' names in year 2008."

Schema_links: [course.title,course.course_id = teaches.course_id,teaches.id = instructor.id,instructor.name,teaches.year,2008]

A: Let's think step by step. For creating the SQL for the given question, we need to join these tables = [course,teaches,instructor].

First, create an intermediate representation, then use it to construct the SQL query.

Intermediate_representation: select course.title , instructor.name from course where teaches.year = 2008 order by course.title asc

SQL: SELECT T1.title , T3.name FROM course AS T1 JOIN teaches AS T2 ON T1.course_id = T2.course_id JOIN instructor AS T3 ON T2.id = T3.id WHERE T2.YEAR = 2008 ORDER BY T1.title

### A.5.3 Nested Complex

# Use the intermediate representation and the schema links to generate the SQL queries for each of the questions.

Table advisor, columns = [*,s_ID,i_ID]
Table classroom, columns = [*,building,room_number,capacity]
Table course, columns = [*,course_id,title,dept_name,credits]
Table department, columns = [*,dept_name,building,budget]
Table instructor, columns = [*,ID,name,dept_name,salary]
Table prereq, columns = [*,course_id,prereq_id]
Table section, columns = [*,course_id,sec_id,semester,year,building,room_number,time_slot_id]
Table student, columns = [*,ID,name,dept_name,tot_cred]
Table takes, columns = [*,ID,course_id,sec_id,semester,year,grade]
Table teaches, columns = [*,ID,course_id,sec_id,semester,year]
Table time_slot, columns = [*,time_slot_id,day,start_hr,start_min,end_hr,end_min]
Foreign_keys = [course.dept_name = department.dept_name,instructor.dept_name = department.dept_name,section.building = classroom.building,section.room_number = classroom.room_number,
section.course_id = course.course_id,teaches.ID = instructor.ID,teaches.course_id = section.course_id,teaches.sec_id = section.sec_id,teaches.semester = section.semester,teaches.year = section.year,student.dept_name = department.dept_name,takes.ID = student.ID,takes.course_id = section.course_id,
takes.sec_id = section.sec_id,takes.semester = section.semester,takes.year = section.year,advisor.s_ID = student.ID,advisor.i_ID = instructor.ID,prereq.prereq_id = course.course_id,prereq.course_id = course.course_id]

Q: "Find the title of courses that have two prerequisites?"

Schema_links: [course.title,course.course_id = prereq.course_id]

A: Let's think step by step. "Find the title of courses that have two prerequisites?" can be solved by knowing the answer to the following sub-question "What are the titles for courses with two prerequisites?".
The SQL query for the sub-question "What are the titles for courses with two prerequisites?" is SELECT T1.title FROM course AS T1 JOIN prereq AS T2 ON T1.course_id = T2.course_id GROUP BY T2.course_id HAVING count(*) = 2
So, the answer to the question "Find the title of courses that have two prerequisites?" is =

Intermediate_representation: select course.title from course where count ( prereq.* ) = 2 group by prereq.course_id

SQL: SELECT T1.title FROM course AS T1 JOIN prereq AS T2 ON T1.course_id = T2.course_id GROUP BY T2.course_id HAVING count(*) = 2

Q: "Find the name and building of the department with the highest budget."

Schema_links: [department.dept_name,department.building,department.budget]

A: Let's think step by step. "Find the name and building of the department with the highest budget." can be solved by knowing the answer to the following sub-question "What is the department name and corresponding building for the department with the greatest budget?".

The SQL query for the sub-question "What is the department name and corresponding building for the department with the greatest budget?" is SELECT dept_name , building FROM department ORDER BY budget DESC LIMIT 1

So, the answer to the question "Find the name and building of the department with the highest budget." is =

Intermediate_representation: select department.dept_name , department.building from department order by department.budget desc limit 1

SQL: SELECT dept_name , building FROM department ORDER BY budget DESC LIMIT 1

Q: "Find the title, credit, and department name of courses that have more than one prerequisites?"

Schema_links: [course.title,course.credits,course.dept_name,course.course_id = prereq.course_id]

A: Let's think step by step. "Find the title, credit, and department name of courses that have more than one prerequisites?" can be solved by knowing the answer to the following sub-question "What is the title, credit value, and department name for courses with more than one prerequisite?".

The SQL query for the sub-question "What is the title, credit value, and department name for courses with more than one prerequisite?" is SELECT T1.title , T1.credits , T1.dept_name FROM course AS T1 JOIN prereq AS T2 ON T1.course_id = T2.course_id GROUP BY T2.course_id HAVING count(*) > 1

So, the answer to the question "Find the name and building of the department with the highest budget." is =

Intermediate_representation: select course.title , course.credits , course.dept_name from course where count ( prereq.* ) > 1 group by prereq.course_id

SQL: SELECT T1.title , T1.credits , T1.dept_name FROM course AS T1 JOIN prereq AS T2 ON T1.course_id = T2.course_id GROUP BY T2.course_id HAVING count(*) > 1

Q: "Give the name and building of the departments with greater than average budget."

Schema_links: [department.dept_name,department.building,department.budget]

A: Let's think step by step. "Give the name and building of the departments with greater than average budget." can be solved by knowing the answer to the following sub-question "What is the average budget of departments?".

The SQL query for the sub-question "What is the average budget of departments?" is SELECT avg(budget) FROM department

So, the answer to the question "Give the name and building of the departments with greater than average budget." is =

Intermediate_representation: select department.dept_name , department.building from department where @.@ > avg ( department.budget )

SQL: SELECT dept_name , building FROM department WHERE budget > (SELECT avg(budget) FROM department)

Q: "Find the id of instructors who taught a class in Fall 2009 but not in Spring 2010."

Schema_links: [teaches.id,teaches.semester,teaches.YEAR,Fall,2009,Spring,2010]

A: Let's think step by step. "Find the id of instructors who taught a class in Fall 2009 but not in Spring 2010." can be solved by knowing the answer to the following sub-question "Find the id of instructors who taught a class in Spring 2010".

The SQL query for the sub-question "Find the id of instructors who taught a class in Spring 2010" is SELECT id FROM teaches WHERE semester = 'Spring' AND YEAR = 2010

So, the answer to the question "Find the id of instructors who taught a class in Fall 2009 but not in Spring 2010." is =

Intermediate_representation: select teaches.ID from teaches where teaches.semester = "Fall" and teaches.year = 2009 and teaches.semester != "Spring" and teaches.year = 2010

SQL: SELECT id FROM teaches WHERE semester = 'Fall' AND YEAR = 2009 EXCEPT SELECT

id FROM teaches WHERE semester = 'Spring' AND YEAR = 2010

Q: "Find the name of the courses that do not have any prerequisite?"
Schema_links: [course.title,course.course_id]
A: Let's think step by step. "Find the name of the courses that do not have any prerequisite?" can be solved by knowing the answer to the following sub-question "What are the courses that have any prerequisite?".
The SQL query for the sub-question "What are the courses that have any prerequisite?" is SELECT course_id FROM prereq
So, the answer to the question "Find the name of the courses that do not have any prerequisite?" is =
Intermediate_representation: select course.title from course where @.@ not in prereq.course_id
SQL: SELECT title FROM course WHERE course_id NOT IN (SELECT course_id FROM prereq)

Q: "Find the salaries of all distinct instructors that are less than the largest salary."
Schema_links: [instructor.salary]
A: Let's think step by step. "Find the salaries of all distinct instructors that are less than the largest salary." can be solved by knowing the answer to the following sub-question "What is the largest salary of instructors".
The SQL query for the sub-question "What is the largest salary of instructors" is SELECT max(salary) FROM instructor
So, the answer to the question "Find the salaries of all distinct instructors that are less than the largest salary." is =
Intermediate_representation: select distinct instructor.salary from instructor where @.@ < max ( instructor.salary )
SQL: SELECT DISTINCT salary FROM instructor WHERE salary < (SELECT max(salary) FROM instructor)

Q: "Find the names of students who have taken any course in the fall semester of year 2003."
Schema_links: [student.id,student.name,takes.id,takes.semester,fall,2003]
A: Let's think step by step. "Find the names of students who have taken any course in the fall semester of year 2003." can be solved by knowing the answer to the following sub-question "Find the students who have taken any course in the fall semester of year 2003.".
The SQL query for the sub-question "Find the students who have taken any course in the fall semester of year 2003." is SELECT id FROM takes WHERE semester = 'Fall' AND YEAR = 2003
So, the answer to the question "Find the names of students who have taken any course in the fall semester of year 2003." is =
Intermediate_representation: select student.name from student where takes.semester = "Fall" and takes.year = 2003
SQL: SELECT name FROM student WHERE id IN (SELECT id FROM takes WHERE semester = 'Fall' AND YEAR = 2003)

Q: "Find the minimum salary for the departments whose average salary is above the average payment of all instructors."
Schema_links: [instructor.salary,instructor.dept_name]
A: Let's think step by step. "Find the minimum salary for the departments whose average salary is above the average payment of all instructors." can be solved by knowing the answer to the following sub-question "What is the average payment of all instructors.".
The SQL query for the sub-question "What is the average payment of all instructors." is SELECT avg(salary) FROM instructor
So, the answer to the question "Find the minimum salary for the departments whose average salary is above the average payment of all instructors." is =
Intermediate_representation: select min(instructor.salary) , instructor.dept_name from instructor where avg ( instructor.salary ) > avg ( instructor.salary ) group by instructor.dept_name
SQL: SELECT min(salary) , dept_name FROM instructor GROUP BY dept_name HAVING avg(salary) > (SELECT avg(salary) FROM instructor)

Q: "What is the course title of the prerequisite of course Mobile Computing?"

 [course.title,course.course_id = prereq.course_id,prereq.prereq_id,course.title,Mobile Computing]

A: Let's think step by step. "What is the course title of the prerequisite of course Mobile Computing?" can be solved by knowing the answer to the following sub-question "What are the ids of the prerequisite of course Mobile Computing?".

The SQL query for the sub-question "What are the ids of the prerequisite of course Mobile Computing?" is SSELECT T1.prereq_id FROM prereq AS T1 JOIN course AS T2 ON T1.course_id = T2.course_id WHERE T2.title = 'Mobile Computing'

So, the answer to the question "What is the course title of the prerequisite of course Mobile Computing?" is =

Intermediate_representation: select course.title from course where @.@ in prereq.* and course.title = "Mobile Computing"

SQL: SELECT title FROM course WHERE course_id IN (SELECT T1.prereq_id FROM prereq AS T1 JOIN course AS T2 ON T1.course_id = T2.course_id WHERE T2.title = 'Mobile Computing')

Q: "Give the title and credits for the course that is taught in the classroom with the greatest capacity."

Schema_links: [classroom.capacity,classroom.building = SECTION.building,classroom.room_number = SECTION.room_number,course.title,course.credits,course.course_id = SECTION.course_id]

A: Let's think step by step. "Give the title and credits for the course that is taught in the classroom with the greatest capacity." can be solved by knowing the answer to the following sub-question "What is the capacity of the largest room?".

The SQL query for the sub-question "What is the capacity of the largest room?" is (SELECT max(capacity) FROM classroom)

So, the answer to the question "Give the title and credits for the course that is taught in the classroom with the greatest capacity." is =

Intermediate_representation: select course.title , course.credits from classroom order by classroom.capacity desc limit 1"

SQL: SELECT T3.title , T3.credits FROM classroom AS T1 JOIN SECTION AS T2 ON T1.building = T2.building AND T1.room_number = T2.room_number JOIN course AS T3 ON T2.course_id = T3.course_id WHERE T1.capacity = (SELECT max(capacity) FROM classroom)

## A.6  Self-correction prompts

### A.6.1  Generic self-correction prompt

The Generic self-correction prompt was implemented in a zero-shot setting, where all queries were assumed to be "Buggy SQL". An example of this prompt is illustrated in Figure 6.

Figure 6: An example of Generic self-correction prompt.

### A.6.2 Gentle self-correction prompt

The Gentle self-correction prompt was implemented in a zero-shot setting. For this self-correction prompt we don't have the assumption of being Buggy and we included some instructions for fixing the SQL queries. An example of this prompt is demonstrated in Figure 7.

**#### For the given question, use the provided tables, columns, foreign_keys, and primary keys to fix the given SQLite SQL QUERY for any issues. If there are any problems, fix them. If there are no issues, return SQLite SQL QUERY as is.**
**#### Use the following instructions for fixing the SQL query:**
**1) Use the database values that are explicitly mentioned in the question**
**2) Pay attention to the columns that are used for the JOIN by using the Foreign_keys.**
**3) Use DESC and DISTINCT when needed**
**4) Pay attention to the columns that are used for the GROUP BY clause.**
**5) Pay attention to the columns that are used for the SELECT clause.**
**6) Only change the GROUP BY clause when necessary.**

**Tables concert, columns = [concert_ID, …]**
**…**
**Foreign_keys = [concert.Stadium_ID = stadium.Stadium_ID, …]**
**Primary_key = [stadium.Stadium_ID, …]**
**#### Question: What is the name and capacity for the stadium with highest average attendance?**
**#### SQLite SQL Query**
**SELECT Name , Capacity FROM stadium ORDER BY Average LIMIT 1**
**#### Fixed SQL QUERY**

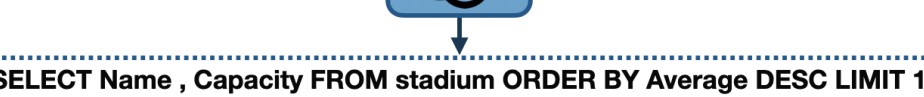

**SELECT Name , Capacity FROM stadium ORDER BY Average DESC LIMIT 1**

Figure 7: An example of Gentle self-correction prompt.