# OpenReview forum: "DIN-SQL: Decomposed In-Context Learning of Text-to-SQL with Self-Correction"
_NeurIPS.cc/2023/Conference — NeurIPS 2023 poster_

### Official Review · Reviewer_Qajd · 2023-06-24

**Soundness:** 4 excellent
**Presentation:** 4 excellent
**Contribution:** 2 fair
**Rating:** 6
**Confidence:** 5

**Summary:**

In their study on the Text2SQL task, the authors demonstrate the efficacy of breaking down the generation problem into sub-problems when utilizing LLMs to improve performance on the Spider dataset. The approach is simple yet effective on the Spider dataset. It yielded consistent improvements across three LLMs: GPT-4, Codex Davinci, and Cushman. Notably, the paper achieves a new SOTA performance on the Spider holdout test set.

**Strengths:**

- SOTA on Spider holdout test set
- Approach consistently improve results on three OpenAI LLMs


**Weaknesses:**

- The analysis and experiments conducted in the study were solely focused on a single dataset, namely Spider.
- The study does not provide sufficient evidence to support the generalizability of the proposed approach to Text2SQL tasks in general.
- It is important to highlight the existence of several large-scale and robust Text2SQL datasets that could have strengthened the results in terms of generalizability. Notably, the BIRD-bench dataset (https://bird-bench.github.io/) provides an additional resource for evaluating the proposed approach. Furthermore, the Spider dataset has various variants, such as Spider-SYN, Dr.Spider, Spider-realistic, and Spider-DK, which offer opportunities to assess the effectiveness of the approach in different contexts and scenarios. Conducting experiments on these datasets would have enhanced the overall robustness and applicability of the findings.
- The foundation models utilized in the study are exclusively based on OpenAI models.
- While the classification step for SQL difficulty proves effective on the Spider dataset, it is unlikely to exhibit strong generalization capabilities to other real-world Text2SQL data. The classification of SQL difficulty may be influenced by annotation artifacts specific to the Spider dataset.


**Questions:**

1. How did the authors come up with the self-correction prompts? Was it through a lot of trial and error from the training and validation set?
2. How did the author decide on this set of in-context learning exemplars?


**Limitations:**

No negative societal impacts

---

> ### Author Rebuttal · Authors · 2023-08-09
>
> The first, second, and third weaknesses mentioned by the reviewer have been thoroughly addressed in the general response to all reviewers.
>
> Regarding the weakness related to other LLMs beside OpenAI models, at the time of writing this paper, GPT-based models, PaLM, and the OPT model were the widely used language models. Access to other LLMs besides the GPT family was limited as they were not accessible through API calls. Moreover, due to the large number of parameters, we faced difficulties in loading those models onto our available infrastructure.
>
> Concerning the last weakness regarding the applicability of our proposed approach to real-world scenarios with more complex SQL query structures, our current classification structure effectively handles numerous benchmarks in the text-to-SQL domain and real-world applications. To accommodate more complex scenarios, we can readily add another class for intermediate steps and include additional steps in the existing most complex class. As the Spider dataset already contains the most complex SQL queries, it serves as a valuable benchmark for assessing intricate SQL structures.
>
>
>
> Question: How did the authors come up with the self-correction prompts? Was it through a lot of trial and error from the training and validation set?
>
> Answer:  The concept of using LLMs to correct themselves was inspired by an application mentioned on the OpenAI website. LLMs were suggested as a tool for debugging codes. Recognizing the LLMs' potential in debugging, we proposed utilizing them to self-correct their own responses. This approach aligns with the ideas presented in independent work conducted around  the same time as ours, such as the paper "self-debug" by Google DeepMind, where LLMs were employed to debug their generated SQL queries and Python codes.
>
> Question: How did the author decide on this set of in-context learning exemplars?
>
> Answer: The examples in our prompts are deliberately selected to include at least one instance for almost all widely used SQL keywords. This careful selection ensures that the LLM has access to all the necessary information before generating the SQL query.

---

> > ### Comment · Reviewer_Qajd · 2023-08-16
> >
> > Thanks for your response and new experiment results. I have updated my ratings accordingly

---

### Official Review · Reviewer_DUDq · 2023-06-29

**Soundness:** 4 excellent
**Presentation:** 4 excellent
**Contribution:** 3 good
**Rating:** 6
**Confidence:** 4

**Summary:**

This paper addresses the task of text-to-SQL prediction using large language model (LLM) prompting. The authors propose DIN-SQL, a chain-of-thought (CoT) prompting method which decomposes the SQL prediction process into four substeps: schema linking, complexity classification, SQL prediction, and self-correction. For each step, a manually designed prompt with few-shot in-context learning samples are provided to the LLM to obtain the prediction. Especially, after complexity classification, samples with different complexity levels are addressed with different prompts. Experiments on the Spider dataset demonstrate the effectiveness of the proposed method.


**Strengths:**

- The overall pipeline is conceptually simple but practically effective. It appears to successfully leveraged the power of LLMs and achieved promising results.

- The idea of applying different prompts for samples with different complexity is justifiable. It is based on prior observations that CoT prompts are effective on harder samples but can hurt the performance on easy samples. The ablation studies also echoed this observation, and verified the usefulness of this proposed mechanism.


**Weaknesses:**

- The proposed method does not carry much techinical innovation, as CoT is already a widely studied and applied technique, and the intermediate representation is imported from previous work NatSQL.

- The complexity categorization and corresponding prompt designs are more or less tailored for the Spider dataset. It might be unclear whether the method is directly applicable in real-world scenarios, where the variety of target SQL might be much larger, and may include additional mechanisms such as self-join.

- Based on my observations on the "sub-questions" generated for nested complex samples, it seems that many of them (though not all) are identical to the original questions. This might not be the desired behavior and leaves room for further improvements. For example:
	- "Find the title, credit, and department name of courses that have more than one prerequisites?" can be solved by knowing the answer to the following sub-question "What is the title, credit value, and department name for courses with more than one prerequisite?".


**Questions:**

- Around L231: The SQL-prediction step prompt for nested complex questions are designed in a comprehensive way, allowing multiple sub-questions to be included. However, looking at the samples in the appendix, it seems that at most one sub-question is predicted. Is the method actually able to handle very complex queries by generating >1 sub-questions?


**Limitations:**

Several limitations are discussed, although I think there are additional points as I mention above.

---

> ### Author Rebuttal · Authors · 2023-08-09
>
> The first weakness raised in the review has been thoroughly addressed in the general response to all reviewers.
>
> Regarding the second weakness concerning the applicability of our proposed approach to real-world scenarios with more complex SQL query structures, our current classification structure is effective for many benchmarks in the text-to-SQL domain, which aim to resemble queries in real-world applications. However, it is fair to say that the queries in real world applications can be more complex. To handle more complex scenarios which fall outside our current query classes, we may simply add more query classes to our query classification module and provide more relevant steps to each class in our query generation module. We could only evaluate our model on existing benchmarks, and Spider, at the time of writing this paper, stood out as the most complex text-to-SQL benchmark we could use.
>
> The third weakness pertains to identical questions and sub-questions. During our experiments with query decomposition, we noticed instances where the query did not have a clear subquery or could not be broken down to independent components. In those cases, the model generated a paraphrased version of the original question. To promote this behavior especially when a question cannot be broken to sub-questions, we included examples of such cases in our prompt. Our evaluation shows that including such cases overall helps the model.
>
> Question: Is the method actually able to handle very complex queries by generating >1 sub-questions?
>
> Answer: The presence of only one sub-question in all the few-shot demonstrations can be attributed to the database selection process from the training set of Spider. For the samples in our chosen database, we did not encounter queries that required more than one sub-question. However, during the inference stage, we noticed that the model was capable of generating multiple sub-questions, even though some were unnecessary. It was evident that often one sub-question, along with a SQL query utilizing JOIN operators, proved to be sufficient.

---

> > ### Comment · Reviewer_DUDq · 2023-08-15
> >
> > Thank you for the response. My overall evaluation of the contributions of the work is not changed, thus I will keep my current score.

---

> > > ### Author Response · Authors · 2023-08-16
> > >
> > > Thank you for your thoughtful comments.
> > > In our effort to address your concerns about the applicability of our approach, we have included the performance of our approach on the BIRD benchmark in the comment section of the rebuttal to all reviewers, where DIN-SQL establishes itself as the state-of-the-art performer. This additional evaluation further underscores the effectiveness of our approach and strengthens its applicability in real-world scenarios. We believe that the results we've presented can provide valuable insights into the reliability and impact of our method.
> > > Your dedication to thoughtful evaluation is invaluable to us, and we're pleased to have been able to provide additional evidence that reinforces the merits of our work. If you have any further questions or observations, please feel free to share them with us.

---

### Official Review · Reviewer_VYLB · 2023-07-04

**Soundness:** 3 good
**Presentation:** 4 excellent
**Contribution:** 2 fair
**Rating:** 5
**Confidence:** 4

**Summary:**

The paper proposed to decompose the text-to-SQL reasoning into multiple steps and solve them with large language models. The authors began by conducting an error study of LLMs with few-shot learning and identified the common errors, such as "schema linking" and "JOIN". To address these common errors, they proposed a new method called DIN-SQL which breaks the text-to-SQL task down into four modules: (1) schema linking, (2) query classification and decomposition, (3) SQL generation, and (4) self-correction.

In the schema linking and query classification modules, examples from the Spider dataset's training set are used to guide LLMs in identifying the relevant database schema and potential SQL structure. The SQL generation modules leverage different examples to handle various SQL structures. For more complex SQL structures, such as non-nested complex and nested-complex queries, demonstration examples with hand-written intermediate reasoning steps are utilized. At last, the self-correction module asks the LLMs to correct the predicted SQL if they realize that their initial generation was incorrect.

The proposed framework demonstrates remarkable performance, surpassing state-of-the-art models on the Spider test set.

**Strengths:**

1. The proposed method DIN-SQL demonstrates very impressive results on the hold-out test set in the Spider dataset, showing the effectiveness of DIN-SQL on Spider.
2. The error analysis reveals the drawbacks of LLMs with standard few-shot demonstrations and the proposed method are designed to address these drawbacks.


**Weaknesses:**

1. The proposed methods seem highly tailored to the examples in the Spider dataset. Previous work [1-2] has highlighted the spurious correlations in the Spider dataset, such as the strong lexical matching between the database and the question. It seems important to evaluate the proposed method on other datasets such as Spider-syn [1], Spider-DK [2], Dr. Spider [3], or KaggleDBQA [4].

2. The proposed SQL generation modules require hand-written intermediate reasoning steps. I don't think it is a problem given the limited number of examples requiring annotation but my concern is whether these examples will still serve as good demonstrations for the datasets other than Spider. I hope this can be addressed along with 1.

[1] Gan, Yujian, et al. "Towards robustness of text-to-SQL models against synonym substitution." arXiv preprint arXiv:2106.01065 (2021).
[2] Deng, Xiang, et al. "Structure-grounded pretraining for text-to-sql." arXiv preprint arXiv:2010.12773 (2020).
[3] Chang, Shuaichen, et al. "Dr. Spider: A diagnostic evaluation benchmark towards text-to-SQL robustness." arXiv preprint arXiv:2301.08881 (2023).
[4] Lee, Chia-Hsuan, Oleksandr Polozov, and Matthew Richardson. "KaggleDBQA: Realistic evaluation of text-to-SQL parsers." arXiv preprint arXiv:2106.11455 (2021).

**Questions:**

I am somewhat confused by the disparity between the results of DIN-SQL in the Spider development set and the Spider test set. On the test set, DIN-SQL demonstrates superior performance compared to the state-of-the-art model [1], achieving a margin of 5.4 points higher (85.3 vs. 79.9). However, on the development set, it falls short by 9.9 points (74.2 vs. 84.1) and is outperformed by other in-context learning methods [2]. Do you know what factors contribute to DIN-SQL's high performance on the test set but comparatively low performance on the development set?

[1] Li, Haoyang, et al. "Resdsql: Decoupling schema linking and skeleton parsing for text-to-sql." Proceedings of the AAAI Conference on Artificial Intelligence. Vol. 37. No. 11. 2023.
[2] Ni, Ansong, et al. "Lever: Learning to verify language-to-code generation with execution." arXiv preprint arXiv:2302.08468 (2023).

**Limitations:**

Looks good to me.

---

> ### Author Rebuttal · Authors · 2023-08-09
>
> The first weakness highlighted by the reviewer has been thoroughly addressed in the general response to all reviewers.
>
> Regarding the second weakness raised in the review, we would like to clarify that our examples were chosen from the training set of Spider and were selected deliberately to include at least one example for each of the widely used SQL keywords. Additionally, the cross-domain nature of the Spider dataset ensures that achieving the highest performance on this benchmark translates to strong performance across various domains. As a result, we believe that the examples used in our prompts are highly applicable to a wide range of existing benchmark datasets.
>
> Question: Why is there a disparity between the results of DIN-SQL in the Spider dev set and the test set?
> Answer: The test set of Spider, which is intended to be hidden, has not been published, making it challenging to determine the exact reason for the performance disparity. However, we have some speculations that may shed light on potential contributing factors:
>
> 1) Our proposed approach relies solely on the database schema to generate answers for given questions. This strategy poses challenges in cases where the model needs knowledge of how values are stored in the database. For instance, when the question is "Give me the name of female users with age over 40," the model may not know whether the stored values in the gender column are "female" or "F." Consequently, it relies on the question's context and chooses "female." We speculate that the number of questions requiring knowledge of specific database values is lower in the test set compared to the dev set of Spider.
>
> 2) Ambiguities exist in the database schema of the dev set of Spider. In some databases within the dev set, tables have columns that store the same information about entities but in a slightly different format. While using any of those columns may be treated as correct, a generated query is deemed correct if its column choice matches that of the reference query. The higher performance of our model on the test set might be attributed to the test set having a more well-structured database schema, reducing such ambiguities.
>
> While these speculations offer insights into potential reasons for the observed performance disparity, the lack of access to the test set hinders us from confirming these hypotheses conclusively.

---

> > ### Comment · Reviewer_VYLB · 2023-08-12
> > **Evaluation on other datasets**
> >
> > Thank you for providing the response.
> > I think that the disparity between the results of DIN-SQL in the Spider dev set and the test set, when compared to other approaches, raises a crucial concern about the robustness/stability of DIN-SQL. Given that, it is essential to evaluate the proposed method on another text-to-SQL dataset.

---

> > > ### Author Response · Authors · 2023-08-16
> > > **DIN-SQL performance on other datasets**
> > >
> > > Thank you for your thoughtful comment. We greatly value your concern about the robustness and stability of the DIN-SQL approach. In response to this concern, we have conducted a comprehensive evaluation of our method on the BIRD benchmark, another challenging text-to-SQL benchmark. We are pleased to share that our evaluation on the BIRD benchmark development and test sets reaffirms the effectiveness and robustness of the DIN-SQL approach.
> > >
> > > We have included the results of our evaluation, where DIN-SQL establishes itself as the state-of-the-art performer, in the comment section of the rebuttal to all reviewers. These results further highlight the consistency and reliability of our approach across different datasets.
> > >
> > > We appreciate your attention to these critical aspects and hope that our evaluation on the BIRD benchmark adds clarity and confidence in the stability of the DIN-SQL method.

---

> > > > ### Comment · Reviewer_VYLB · 2023-08-21
> > > >
> > > > Thank the authors for answering my question. I prefer the paper to be accepted.

---

### Official Review · Reviewer_XH1V · 2023-07-05

**Soundness:** 3 good
**Presentation:** 3 good
**Contribution:** 2 fair
**Rating:** 4
**Confidence:** 4

**Summary:**

This paper proposes to improve few-shot prompting LLMs for text-to-SQL task. It first provides detailed error analysis on existing few-shot prompting LLM approaches, into six categories. Then the paper proposes a new approach to decompose the task into a few sub-tasks, solve each task individually, and compose the subtask for the final answer.

Experiments show the proposed approach achieve state-of-the-art on the Spider benchmark dataset.

**Strengths:**

1. The result is solid, as it achieves state-of-the-art on the challenging Spider benchmark dataset. In addition, this paper provides detailed ablation study and analysis, therefore it's straightforward to understand the strength and weakness of the proposed approach.
2. The paper is well-written, easy to follow the motivation and details.

**Weaknesses:**

Overall, my biggest concern is there's little additional novelty beyond "just another prompt engineering paper". It's expected that with larger LM, few-shot approaches could outperform existing fine-tuning based approaches. However, after reading this paper, it's still unclear whether scaling LLM can solve this text-to-SQL task and if not, where is the bottleneck.

More specific concerns:
1. The manual error analysis (Section 3) only applies to Codex, it's unclear whether these errors still exist for larger language models such as GPT4.
2. There's only marginal improvement compared with chain-of-thought prompting (Table 4). And I guess the margin would be even smaller if  we use GPT4 as LLM.


**Questions:**

1. What's the difference on errors for Codex and GPT4
2. What's the result on chain-of-thought promoting for GPT4.
3. What do you think is the upper bound for this dataset, and do you think scaling LLM can achieve the upper bound?

---

> ### Author Rebuttal · Authors · 2023-08-09
>
> A detailed explanation of the novelties of our prompting method is provided in the general response to the reviewers.
> The argument regarding marginal improvement over the Chain-of-thought method is invalid because the cited performance “decomposed COT prompting” is not for the chain-of-thought method alone. The reported performance here is for our comprehensive framework that includes all our components except classification. This means that the approach contains not only COT prompting but also Schema Linking, Self-Correction, and NatSQL Intermediate Representation, all of which are significant contributions of our work. It is essential to consider the entire framework's performance, as each module contributes to the overall effectiveness of our approach.
>
>
> Question: What's the difference on errors for Codex and GPT4
>
> Answer: Our error analysis reveals that the majority of errors in the area of schema-linking remains in transitioning from Codex to GPT-4. Ambiguities within the database schema pose a significant challenge, even for advanced models like GPT-4, in accurately identifying correct columns and tables. While using a larger model addresses certain issues, particularly in cases where the model struggles to generate accurate SQL queries (miscellaneous class of errors), schema-linking complexities remain the primary obstacle.
>
> In summary, larger models generally exhibit enhanced SQL query generation capabilities, yet the challenges arising from schema-linking ambiguities remain prominent. Addressing these challenges will be a key focus for further improvement and refinement of our approach.
>
>
> Question: What's the result of chain-of-thought promoting for GPT4.
>
> Answer: We did not specifically test the pure chain-of-thought approach on the Spider dataset with GPT-4. However, we evaluated the performance of the chain-of-thought approach when integrated with three other modules proposed by us. Using the chain-of-thought approach uniformly for all questions, irrespective of their complexity, resulted in a performance degradation, as highlighted in Table 4. The decomposed chain-of-thought result presented in this table refers to employing the most complex prompt, developed for the nested complex class, for all questions instead of adopting a classification-based approach to determine prompt complexity based on the question's level of difficulty.
>
> Question: What do you think is the upper bound for this dataset, and do you think scaling LLM can achieve the upper bound?
>
> Answer: We think we are approaching the upper bound on the Spider dataset because there are certain ambiguities in the database, which make it challenging to achieve a flawless performance even if larger LLMs are employed. Some natural language questions have multiple interpretations, and relying on a single reference query as the gold standard is insufficient to address these diverse interpretations. While scaling LLMs can address some issues related to generating complex SQL queries, the primary challenge persists in the area of schema linking, where comprehending database schema based on less-descriptive table and column names remains a difficult task.

---

> > ### Comment · Area_Chair_TxWL · 2023-08-18
> > **Please Reply to Author Rebuttal**
> >
> > Dear reviewer,
> >
> > Thanks a lot for your efforts and valuable reviews. Would you please check this author rebuttal and see how they address your concerns on novelty, manual error analysis, and marginal improvement? Please reply to authors by adding your following comments below this author rebuttal.
> >
> > As the author-reviewer discussion is closed soon, we would appreciate if you could submit your reply to authors by Aug 21st 1pm EDT.
> >
> > Thanks!
> >
> > Best,
> > AC

---

> > > ### Comment · Reviewer_XH1V · 2023-08-21
> > >
> > > I have read other reviews and the rebuttal carefully. I stand with my point that there's marginal improvement over COT prompting and will not change my score.

---

### Official Review · Reviewer_kanX · 2023-07-08

**Soundness:** 3 good
**Presentation:** 3 good
**Contribution:** 3 good
**Rating:** 6
**Confidence:** 4

**Summary:**

The paper's contributions are the following:
- The authors examined the common failure modes of doing text-to-SQL with few-shot prompting of LLMs: schema linking, JOIN, GROUP BY, nested queries and set operations, invalid SQL and miscellaneous.
- The authors propose a method to do text-to-SQL by decomposing the task into 4 sub-problems, and solving each with a few-shot / zero-shot prompt to an LLM: schema linking, classification and decomposition, SQL generation, and self-correction. Their in-context learning method is reported to attain the highest execution accuracy on the test set of the Spider dataset, without making use of database content.
- The work shows that LLMs can be used for text-to-SQL via prompting with performance equivalent to or better than methods that make use of fine-tuning.


**Strengths:**

- The proposed method is a good application of chain-of-thought style problem decomposition to in-context learning techniques for text-to-SQL.
- The method is well motivated by first conducting an examination of common failure modes when using LLMs for text-to-SQL in the few-shot setting. Investigating the error improvements using their method in Figure 4 is a very clear way to show how their method helps with better in-context learning.
- The paper is written in a largely clear manner that makes it easy to follow.
- The paper's results are a relevant contribution to the text-to-SQL community, in that their performance is significantly better than other LLM techniques like those in Rajkumar et al., 2022 and Liu et al., 2023.

**Weaknesses:**

- There could have been more discussion about how the proposed four modules have been implemented without using prompting, to better situate the work in the literature.
- The paper should have been clearer about how the intermediate representation of NatSQL bridges the gap between queries and SQL statements. An example each for the non-nested complex class and nested complex class would have been helpful in the main paper, instead of leaving it to the Appendix. In particular, it is not clear in the paper 1) how the intermediate representation is obtained, 2) how the removal of operators like GROUP BY or the WHERE clause from the syntax of the intermediate representation can help the LLM still generate the right SQL statements and 3) how the LLM is induced to solve sub-queries for the nested complex class.
- The paper could have been clear about the latency of its proposed method. While high-performing, it is likely that making several sequential calls to an LLM will be high latency.


**Questions:**

- How is the intermediate representation obtained? I refer to Appendix A.5.2 to see that it is inserted into the prompt, but it is not clear where this is generated.
- How do the authors think the removal of operators like GROUP BY or the WHERE clause from the syntax of the intermediate representation help the LLM still generate the right SQL statements?
- Can the authors illustrate how a query is decomposed into sub-queries for the nested complex class? It is not clear to me still in Appendix A.5.3.
- What is the latency of the proposed decomposed in-context learning method, and how does it compare to other methods (like RESDSQL-3B + NatSQL or Graphix-3B+PICARD)?

**Limitations:**

Yes, limitations are adequately addressed

---

> ### Author Rebuttal · Authors · 2023-08-09
>
> Question: How is the intermediate representation obtained (ref Appendix A.5.2)?
>
> Answer: For our few-shot examples, we used the intermediate representation from the NatSQL Github repo. The repo gives the intermediate representation for all queries in the training set of Spider.
>
> Question: How does removing operators in the intermediate representation help?
>
> Answer: Removing/merging operators makes the transition from natural language to SQL easier and is part of our problem decomposition. Expressions in natural language queries may not clearly map to a unique SQL clause or they may map to multiple clauses. For example, some conditions are mapped to the WHERE clause whereas others are mapped to the HAVING clause. Some SQL clauses do not have a clear counterpart in text descriptions (e.g. JOIN and GROUP BY). Dispensing or merging the operators makes the generation task easier and pushes the LLM to focus on correctly predicting the query structure before refining it in the next step.
>
> Question: Can the authors illustrate how a query is decomposed into sub-queries for the nested complex class? It is not clear to me still in Appendix A.5.3.
>
> Answer: The sub-questions are extracted in the classification and decomposition module. For example, as shown in Fig 3-b, for the question “how many courses that do not have prerequisites,” the sub-question: “which courses have prerequisites” is extracted. In SQL generation, SQL is generated for sub-questions in a step-by-step reasoning process before generating the whole query. For example, consider the last demonstration in the prompt in Appendix A.5.3, where the question “Give the title and credits for the course that is taught in the classroom with the greatest capacity” has the sub-question “What is the capacity of the largest room?”. The mapping of the sub-question to SQL, i.e. “ (SELECT max(capacity) FROM classroom)”, is provided in the reasoning process before generating the final answer.
>
> Question: Latency compared to other methods?
>
> Answer: Our latency heavily relies on that of the OpenAI API calls which varies a lot over time. For GPT-4, this was typically under 1min. This time can significantly be improved using the LLM on Microsoft Azure. The latency of other approaches also depends heavily on the hardware and the GPU.

---

> > ### Comment · Area_Chair_TxWL · 2023-08-18
> > **Please Reply to Author Rebuttal**
> >
> > Dear reviewer,
> >
> > Thanks a lot for your efforts and valuable reviews. Would you please check this author rebuttal and see how they address your concerns on more discussion on implementation without prompting, intermediate representation of NatSQL, and latency of its proposed method? Please reply to authors by adding your following comments below this author rebuttal.
> >
> > As the author-reviewer discussion is closed soon, we would appreciate if you could submit your reply to authors by Aug 21st 1pm EDT.
> >
> > Thanks!
> >
> > Best,
> > AC

---

> > ### Comment · Reviewer_kanX · 2023-08-21
> >
> > Thank you for your reply and the encouraging results on the BIRD dataset. I have read the other comments and will maintain my rating.

---

### Author Rebuttal · Authors · 2023-08-09

Our method draws inspiration from chain-of-thought and decomposed prompting techniques and brings valuable contributions to prompting across various domains, including text-to-SQL. These contributions are as follows:
1) Adaptive Prompting Based on Task Complexity: Our technique involves classifying the input task complexity and adjusting the prompt complexity accordingly. Tailoring prompts based on input question complexity outperforms using generic or overly complex prompts, as demonstrated in Table 4. Moreover, this classification based on task complexity can reduce the number of tokens that are used for simple questions, hence reducing the cost.
2) Schema-Linking Module: Our work introduces a schema-linking module, which has not been utilized in the context of prompting approaches. Inspired by our work, a few other prompting methods (e.g. C3 paper and LangChain SQLAgent) boost their performance using schema linking. The schema-linking template we propose is optimized through numerous iterations to mimic human thought processes.
3) LLMs for Self-Correction: Our research is among the first to propose using Language Models (LLMs) to self-correct their generated responses. The effectiveness of this method is demonstrated by other independent work conducted around the same time as ours, including the self-debug paper, across various domains, not limited to text-to-SQL. Our approach influenced the widely used text-to-SQL agent of LangChain, with practical applications in the industry.
In addition to the aforementioned contributions in the prompting techniques domain, our method stands out with the highest performance, surpassing not only fine-tuning approaches but also other prompting methods.

For our evaluation, we utilized the Spider dataset, a comprehensive cross-domain benchmark with databases from various domains. Given its extensive coverage, a successful approach on this dataset is expected to perform well across all domains. There are also a few other datasets that are derived from or are based on Spider:
1) Spider-Syn replaces question terms in Spider with synonyms. As reported in the Spider-Syn paper, 99% of those modifications are done on Schema words and the remaining 1% on cell values.
2) Spider-DK modifies the questions in Spider by adding domain knowledge or question paraphrases. For example, "in the order of birth date" in Spider is replaced with "order of their birth date from old to young" in Spider-DK, and "dog" is replaced with "abandoned dogs".
3) Spider-realistic modifies the questions in the Spider dataset to remove the explicit mentions of column names.
4) Dr.Spider applies perturbations to Spider tabases, natural language questions, and SQL queries to measure the robustness of the models.

We did not evaluate our work on these variants for two reasons:
1) Many of these benchmarks share a similar SQL query structure with the Spider dataset but extend the benchmark in one direction. Based on our experience with these datasets, the modified queries in these variants tend to be less complex, and our classification and decomposition techniques are expected to do well on these variants. Also, with the wealth of information stored within the parameters of LLMs, and the less reliance of our method on the training data, our model is expected to do well on these variants compared to fine-tuned models that heavily rely on the training data. For example, LLMs have demonstrated exceptional paraphrasing capabilities, making them resilient to synonym substitutions. Experimental results in the paper "A comprehensive evaluation of ChatGPT’s zero-shot Text-to-SQL capability" support these claims, as the model's performance remained stable under various perturbations to the natural language questions.
2) Cost was the main factor in evaluating our model. We had to use the OpenAI API calls in our evaluation, and the cost of making those API calls with the large number of queries in the Spider dataset was not cheap. Running our model on other variants would have doubled or tripled the cost.

Although we could not provide results for these other datasets due to the expensive nature of employing GPT-4, the resilience of the previous prompting method on these variants suggests a similar trend to hold for our method.

BIRD, another cross-domain dataset similar to Spider, has been very recently proposed.  At the time of writing this paper, the BIRD dataset was not published, hence it was not included in our study. This dataset intentionally focuses on database values and external knowledge provided in queries, whereas our queries are generated independent of database values and only based on database schema for generality reasons (as discussed in the paper). Our prompts for this dataset will need to include such information to be effective. We are currently working with the authors of BIRD to evaluate our model on their datasets.

---

> ### Comment · Reviewer_Qajd · 2023-08-12
> **Spider variant or other text2sql datasets**
>
> Thank you for your response.
>
> > 1. Many of these benchmarks share a similar SQL query structure with the Spider dataset but extend the benchmark in one direction.
> > 2. Cost was the main factor in evaluating our model. We had to use the OpenAI API calls in our evaluation, and the cost of making those API calls with the large number of queries in the Spider dataset was not cheap. Running our model on other variants would have doubled or tripled the cost.
>
> Given the significant cost implications associated with evaluating a few thousands of examples, I remain skeptical about the contribution and practicality of this method for real-world natural language interfaces to databases. I think both BIRD and Dr. Spider evaluate systems from multiple directions.
>
> Also, there are non-API open-source LLMs such as Llama and GPT-J-6B.

---

> > ### Author Response · Authors · 2023-08-16
> > **generalizability of our approach and the choice of LLMs**
> >
> > Thank you so much for your thoughtful review.
> > 1. We have evaluated the performance of our method on the BIRD dataset, as reported in a separate responses to all reviewers. Our performance evaluations on both the development and test sets of both the Spider and BIRD datasets underscore the effectiveness and the generalizability of our proposed approach.
> > 2. Regarding the choice of LLMs, prompt engineering has emerged as a prominent feature of large language models (LLMs), and the effectiveness of most prompting methods is generally observed in models exceeding a certain scale. GPT-J-6B, being a relatively compact model, does not exhibit great performance with many prompting strategies. The alternative LLM family suggested, including Llama, while noteworthy, does not align seamlessly with code generation. As outlined in the paper titled “CodeT5+: Open Code Large Language Models for Code Understanding and Generation,” even the most robust Llama models fall short in comparison to Codex, GPT-3-5-turbo, and GPT-4 in terms of performance. Thus, to genuinely showcase the capabilities of our proposed method, we have chosen to leverage the GPT-based LLM family.

---

> ### Author Response · Authors · 2023-08-16
> **DIN-SQL's triumph on BIRD benchmark**
>
> As a follow-up on the applicability of our approach to other text-to-SQL benchmarks, we now have the results on BIRD, another challenging benchmark that was released after our paper and was cited by reviewers. Based on a comprehensive evaluation conducted jointly with the authors of BIRD, we are happy to announce that our approach not only demonstrates outstanding performance on their benchmark, but it also sets a new state-of-the-art performance.
> The final performance metrics for our approach are as follows: 50.72 in Execution Accuracy (EX) and 58.79 in Valid Efficiency Accuracy (VES) on the development set; and 55.90 in EX and 59.44 in VES on the test set. These results are remarkable, truly confirming the robustness and effectiveness of our method across different databases and benchmarks.
> For comparison, the prior state-of-the-art methodologies achieved scores of 46.35 in EX and 49.77 in VES on the development set, along with 54.89 in EX and 60.77 in VES on the test set. It should be noted that Spider queries cover 200 different databases and BIRD queries cover 95 different databases.
> These outcomes not only validate the efficacy of our approach but also highlight its potential for generalization to diverse text-to-SQL scenarios.

---

> > ### Comment · Reviewer_VYLB · 2023-08-18
> >
> > Thank you for providing the experiment. I think it demonstrates the robustness of DIN-SQL. One follow-up question about the BIRD results: does other methods (e.g. GPT-4) on the leaderboard (https://bird-bench.github.io/) is under a zero-shot or few-shot setting? How does GPT-4 + DIN-SQL perform compared to GPT-4 zero-shot or GPT-4 + few-shot examples from Spider-train?

---

> > > ### Author Response · Authors · 2023-08-18
> > >
> > > Thank you for your insightful feedback. Your attention to detail is greatly appreciated.
> > >
> > > Regarding the performance of GPT-4 in the BIRD leaderboard, we took the initiative to reach out to the authors for clarification on their methodology. Their response confirmed that the prompts they employed for evaluation closely align with those outlined in their paper. Our understanding is that the models marked without (COT) on the leaderboard correspond to zero-shot and models marked with (COT) indicate instances where one pseudo-example is combined with a zero-shot COT approach (Let’s think step by step).
> > >
> > > To address your query regarding a comparative analysis between our method and both few-shot and zero-shot approaches, our paper incorporates a comprehensive experiment conducted on the Spider development set. As demonstrated in Table 3, our approach, DIN-SQL in conjunction with GPT-4, achieved an execution accuracy of 74.2. In contrast, the zero-shot and few-shot performances using GPT-4 alone attained success rates of 64.9 and 67.4, respectively. These results unequivocally showcase the notable performance enhancement achieved by our method over both zero-shot and few-shot scenarios.

---

### Decision · Program_Chairs · 2023-09-21

**Decision:**

Accept (poster)

**Comment:**

This paper presents a group of techniques using few-shot prompting LLMs for text-to-SQL by decomposing the task into smaller sub-problems including schema linking, query classification and decomposition, SQL generation, and self-correction. It achieves state-of-the-art execution accuracy on Spider, a well-studied established benchmark for text-to-SQL. While reviewers pointed out that this is a combination of known techniques for a single task with limited novelty and insights for other problems, it shows the great potential of using LLMs with prompting to solve text-to-SQL tasks better than specialized fine-tuning models. During rebuttal, the author further provided results on BIRD-SQL, a more recent and realistic text-to-SQL benchmark, to dismiss reviewers' concerns on its generalizability and usability when evaluating only on Spider, which is appreciated by reviewers.